# MARS-SQL: A MULTI-AGENT REINFORCE-MENT LEARNING FRAMEWORK FOR TEXT-TO-SQL

## ABSTRACT

Translating natural language to SQL remains difficult for complex queries. Such queries often need environmental interaction and self-correction. To address this, we introduce **MARS-SQL**, a novel multi-agent framework that combines principled task decomposition and interactive reinforcement learning (RL). Our system comprises three specialized agents: a Grounding Agent for schema linking, a Generation Agent for query generation, and a Validation Agent for final selection. The core of our framework is the Generation agent, which is trained via a multi-turn RL policy. Adopting a ReAct-style Think-Act-Observe loop, the agent iteratively generates thoughts, executes SQL actions against a live database, and revises its strategy based on execution feedback, enabling dynamic, stateful reasoning and self-correction. At inference time, we generate multiple interaction trajectories to explore diverse reasoning paths. The Validation agent, then selects the optimal trajectory by modeling verification as a next-token prediction task and choosing the solution with the highest generation probability. This structured workflow pipelines specialized agents. It combines interactive RL for generation with generative modeling for verification. The approach proves highly effective for robust and accurate SQL generation. Experiments show that **MARS-SQL** achieves state-of-the-art Execution Accuracy of 77.84% on the BIRD dev set and 89.75% on the Spider test set.

## 1 INTRODUCTION

Translating natural language questions into executable Structured Query Language (SQL) is an essential task that allows non-expert users to access structured data (Xie et al., 2025a; Li et al., 2024a; 2023). Recent Large Language Models (LLMs) can generate simple queries for well-organised academic benchmarks. However, they often struggle with the complexity of real-world enterprise databases (Hong et al., 2025; Lei et al., 2025). To bridge this gap and tackle the challenges of interacting with complex, real-world databases, researchers have started developing SQL agents (Li et al., 2025b; Wang et al., 2025b; Li et al., 2025c). Instead of producing a query in one step, an SQL agent allows an LLM to interact with the database through multiple rounds of reasoning and feedback. This interactive process resembles how human analysts explore data, making it a more natural and effective way to handle complex database tasks.

Current methodologies in the broader field of AI agents have explored several distinct avenues. A prominent strategy is the use of multi-agent systems, where a complex task is decomposed into specialized sub-tasks, each handled by a dedicated agent (Chang et al., 2024; Huang et al., 2025a; Hong et al., 2024). A parallel line of work uses test-time scaling methods that generate multiple candidate queries and then select the best one (Ni et al., 2023; Li et al., 2022). In the specific domain of Text-to-SQL, these methodologies manifest in two primary forms. One approach relies on monolithic models, which handle schema comprehension, logical planning, and SQL generation in a single pass (Pourreza et al., 2025; Li et al., 2024b). Another prominent approach involves multi-agent frameworks that improve modularity by using API calls to closed-source LLMs, where different agent roles are defined mainly through prompting (Pourreza et al., 2024; Liu et al., 2025b).

At first glance, SQL agents appear to be a straightforward solution. However, the disparity between human intuition and current LLM reasoning leads to significant limitations in their practical application. These challenges include (i) **Compositional reasoning:** Agents often struggle to formulate and maintain a coherent long-term plan required for complex queries. They may fail to correctly combine multiple SQL clauses—such as joins, subqueries, and aggregations—often getting stuck in a loop of fixing minor syntax without addressing the flawed high-level logic (Chaturvedi et al., 2025). (ii) **Schema understanding:** When faced with a large and noisy schema, an agent's exploration can be inefficient. It may repeatedly attempt to query hallucinated columns or fail to identify the correct join keys, leading to multiple turns of unproductive interactions with the database (Deng et al., 2025). (iii) **Environmental grounding:** While interactivity is central to the agent concept, current models often lack the nuanced ability to fully leverage environmental feedback (Huang et al., 2025b). They struggle to diagnose specific SQL dialect errors or recover from ambiguous execution outcomes, limiting their self-correction capabilities (Zhang et al., 2025a). The confluence of these challenges in compositional reasoning, schema understanding, and environmental grounding presents a significant cognitive load that is difficult for any single agent to manage effectively. We, therefore, posit that a multi-stage approach is essential to systematically address these issues.

To overcome these limitations, we introduce **MARS-SQL**, a novel framework built on a multi-stage methodology. This approach has a dual meaning: (1) a multi-agent architecture for principled task decomposition, and (2) a multi-turn reasoning process for interactive query construction. As we highlight in Table 1, our approach integrates key capabilities, such as interactive reasoning and multi-agent collaboration, that are largely absent in existing open and closed-source systems. Our multi-agent system divides the labor across three specialized agents: a **Grounding Agent** for reasoning-driven schema identification, a **Generation Agent** for Multi-turn Trajectory Generation, and a **Validation Agent** for Verification and Selection, allowing each to excel at its sub-task. The core innovation of our framework is the Generation Agent's multi-turn reasoning, which is trained via an interactive reinforcement learning (RL) policy. Adopting a ReAct-style Think-Act-Observe loop (Yao et al., 2022), the agent iteratively builds the query by generating thoughts, executing partial SQL actions against the live database, and observing the feedback. As shown in Figure 1, this interactive process enables dynamic reasoning and self-correction based on real-world outcomes. To further enhance robustness, we generate multiple interaction trajectories at inference time, and the Validation Agent

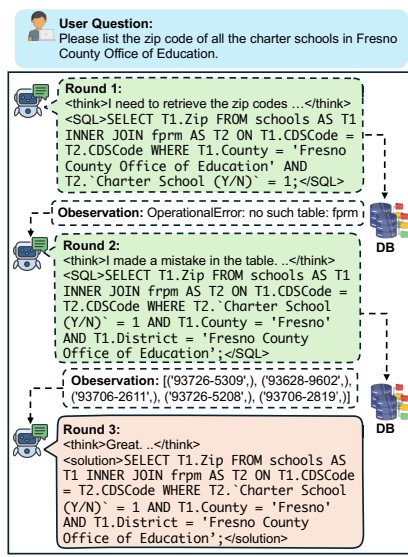

Figure 1: Illustration of interactive reasoning process in **MARS-SQL**.

efficiently selects the optimal solution. This multi-agent workflow proves highly effective, with **MARS-SQL** achieving a new state-of-the-art Execution Accuracy of 77.84% on the BIRD development set and 89.75% on the Spider test set, surpassing previous leading methods.

Our main contributions are summarized as: (1) We introduce **MARS-SQL**, a novel multi-agent framework that tackles complex Text-to-SQL problems through task decomposition and specialized agent training. (2) We propose a

Table 1: Feature Comparison of Text-to-SQL Frameworks.

| Capability | Open-Source | Closed-Source | MARS-SQL |
|---|---|---|---|
| Multi-Agent Architecture | ✗ | ✓ | ✓ |
| Complex Schema Support | ✗ | ✓ | ✓ |
| Interactive Reasoning | ✗ | ✗ | ✓ |
| Efficient Selection | ✗ | ✗ | ✓ |

stateful, interactive SQL generation process, structured as a multi-turn reinforcement learning (RL) policy within a Think–Act–Observe loop, which leverages live database feedback for dynamic reasoning and self-correction. (3) We design a complementary agent workflow that combines a Grounder for schema linking, a Generator for interactive query construction, and a Verifier that reframes candidate selection as a next-token prediction task, yielding a robust mechanism for identifying the optimal solution. (4) We demonstrate state-of-the-art performance, with **MARS-SQL** achieving an execution

accuracy of 77.84% on the BIRD development set and 89.75% on the Spider test set, highlighting the effectiveness of our interactive, multi-agent approach.

## 2 PRELIMINARIES

**Background Formulation.** The primary goal of a Text-to-SQL system is to translate a natural language question into an executable SQL query. We can formally define this task as learning a mapping from a user question and a group of database schemas to the corresponding SQL query.

Let $Q$ be the natural language question posed by a user. Let $S$ be the database schema, which defines the structure of the database. The schema $S$ consists of a set of tables $T = \{t_1, t_2, \ldots, t_m\}$, where each table $t_i$ is composed of a set of columns $C_i = \{c_{i,1}, c_{i,2}, \ldots, c_{i,k}\}$. The schema also includes information about data types, primary keys (PKs), and foreign keys (FKs) that define the relationships between tables. The objective is to generate a SQL query $Y$ such that when it is executed on the database instance $D$, it produces the correct answer to the question $Q$.

Conventionally, the Text-to-SQL problem is treated as a sequence-to-sequence translation task, where the goal is to learn a function $f$:

$$Y = f(Q, S) \tag{1}$$

This formulation, however, treats the generation as a single, static step and fails to capture the exploratory and corrective nature required for solving complex analytical queries.

**Reformulation as an Interactive Decision Process.** As highlighted in the introduction, the static, one-shot formulation is insufficient for complex reasoning. A human analyst does not simply translate; they interact, explore, and refine. To model this more robust process, we reformulate Text-to-SQL as a sequential decision-making task, grounded in the ReAct paradigm (Yao et al., 2023).

Instead of learning a direct mapping to a final query, our goal is to learn an optimal **policy**, $\pi$, that generates a **trajectory** of thoughts and actions to solve the problem. A complete interaction trajectory, $\tau$, is a sequence of multiple rounds:

$$\tau = (h_1, \alpha_1, \omega_1, \ldots, h_M, \alpha_M, \omega_M) \tag{2}$$

Each turn in the trajectory consists of:

- **Thought** ($h_t$): An internal reasoning step where the agent analyzes the problem state, reflects on past observations, and plans the next action.
- **Action** ($\alpha_t$): An operation chosen by the agent from a predefined action space $\mathcal{A}$. In our framework, this primarily involves executing SQL queries against the database.
- **Observation** ($\omega_t$): The feedback received from the environment after executing action $\alpha_t$. This could be a query result, a database error, or other information that guides the agent's next thought.

Under this formulation, the objective is to learn an optimal policy $\pi(\alpha_t | Q, S, (h_{<t}, \alpha_{<t}, \omega_{<t}))$ that maximizes the expected total reward over the trajectory, $E[R(\tau)]$. The reward $R(\tau)$ is typically determined by the final outcome—whether the trajectory successfully produces a correct and executable SQL query. This interactive, policy-based formulation naturally accommodates the trial-and-error and self-correction that are essential for tackling complex, real-world database queries.

## 3 METHODOLOGY

As illustrated in Figure 2, we introduce **MARS-SQL**, a novel multi-agent framework that treats Text-to-SQL generation as an interactive, tool-augmented decision-making process. The framework operates in three stages: Grounding, Generation, and Validation. Initially, a Grounding Agent prunes the full database schema to only the tables and columns relevant to the user question. Subsequently, a Generation agent executes a multi-turn rollout, producing multiple distinct interaction trajectories by actively querying the database. Finally, a Validation Agent scores each trajectory, and the one with the highest confidence score is selected as the final answer.

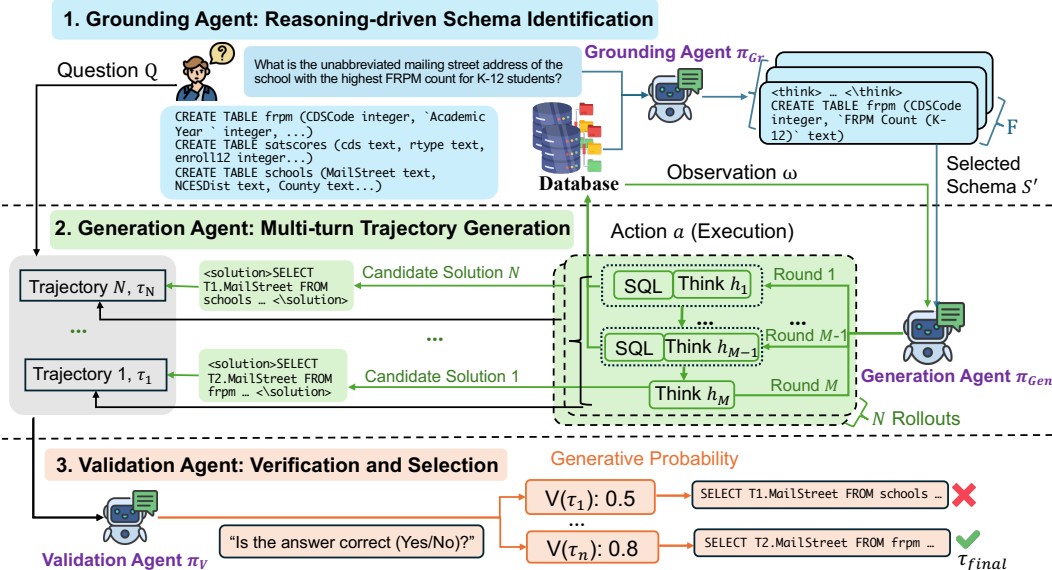

Figure 2: The three-stage workflow of MARS-SQL. (1) Grounding: A Grounding Agent selects the relevant schema. (2) Generation: A Generator agent produces multiple interaction trajectories using a Think-Act-Observe loop. (3) Validation: A Verifier agent scores and selects the best trajectory.

## 3.1 GROUNDING AGENT: REASONING-DRIVEN SCHEMA IDENTIFICATION

The Grounding Agent performs table-level schema linking. Its goal is to learn a policy $\pi_{Ground}$. For each table $t_i \in T (1 \le i \le F)$ and the user question $Q$, the agent takes the pair $x_i = (Q, t_i)$ as input. It then generates a structured output $o_i = (d_i, C_i')$, where $d_i \in \{\text{'Y', 'N'}\}$ is the relevance decision and $C_i' \subseteq C_i$ is the predicted subset of essential columns. The final output of this stage is the reduced schema $S'$, containing only the tables and columns deemed relevant: $S' = \{(t_i, C_i') \mid o_i \text{ has } d_i = \text{'Y'}\}$.

**Training Algorithm.** We train the agent using **Group Relative Policy Optimization (GRPO)** (Shao et al., 2024). For each input $x_i$, the model generates a group of $G$ candidate outputs $\{o_1, \dots, o_G\}$. The policy $\pi_\theta$ is then updated via the objective:

$$J_{\text{GRPO}}(\theta) = \mathbb{E} \left[ \frac{1}{G} \sum_{j=1}^{G} \min \left( \frac{\pi_\theta(o_j|x_i)}{\pi_{\theta_{old}}(o_j|x_i)} A_j, \text{clip} \left( \frac{\pi_\theta(o_j|x_i)}{\pi_{\theta_{old}}(o_j|x_i)}, 1 - \epsilon, 1 + \epsilon \right) A_j \right) - \beta D_{\text{KL}}(\pi_\theta \| \pi_{\text{ref}}) \right]$$

(3)

where $A_j$ is the advantage for candidate $o_j$. The agent's prompt template is in Appendix 14.

**Reward Design.** The reward function $R_{Ground}$ provides a granular score based on the accuracy of the agent's prediction. Let the agent's parsed prediction be $P = (d_p, C_p)$, where $d_p \in \{\text{'Y', 'N'}\}$ is the relevance decision and $C_p$ is the set of predicted columns. Let the ground truth be $o^* = (d_g, C_g)$. The reward $R_g(o, o^*)$ is defined as:

$$R_{Ground}(o, o^*) = \begin{cases} 1.0 & \text{if } o = o^* \text{ (perfect match)} \\ \max(0.5, \frac{|C_g|}{|C_p|}) & \text{if } d_p = d_g = \text{'Y' and } C_g \subset C_p \text{ (superset)} \\ 0.2 & \text{if } d_p = \text{'Y' and } d_g = \text{'N' (incorrect 'Y')} \\ 0.1 & \text{if } d_p = d_g = \text{'Y' and } C_g \not\subseteq C_p \text{ (missing columns)} \\ 0.0 & \text{if response format is invalid} \end{cases}$$

This scheme rewards perfect accuracy while providing partial credit for nearly correct answers, guiding the agent towards effective schema linking.

## 3.2 GENERATION AGENT: MULTI-TURN TRAJECTORY GENERATION

The Generation Agent is the central component, tasked with producing SQL queries. Its **input** is the user question $Q$ and the reduced schema $S'$ from the Grounding Agent. Its **output** is a set of $N$ candidate interaction trajectories, $\{\tau_1, \ldots, \tau_N\}$, where each trajectory comprises of $M$ rounds of the Think-Act-Observe process. The correct trajectory is expected to result in the final SQL solution $Y_i$.

**MDP Formulation.** We model the multi-turn generation process as a Markov Decision Process (MDP), defined by the tuple $(\mathcal{S}, \mathcal{A}, P, R)$.

- **State Space** $\mathcal{S}$: A state $s_t$ represents the history of interaction up to round $t$, containing the sequence of past thoughts, actions, and observations $((h_1, \alpha_1, \omega_1), \ldots, (h_{t-1}, \alpha_{t-1}, \omega_{t-1}))$.

- **Action Space** $\mathcal{A}$: An action $a_t = (h_t, \alpha_t)$ consists of generating a thought $h_t$ and an executable SQL snippet $\alpha_t$.

- **Transition** $P$: $P(s_{t+1}|s_t, a_t)$ is the transition probability, which is determined by the environment (i.e., the database executing the action $\alpha_t$).

- **Reward** $R$: The reward function $R_{gen}(\tau)$ provides a sparse signal based on the final outcome of a complete trajectory $\tau$.

The goal is to learn a policy $\pi_{Gen}(a_t|s_t)$ that maximizes the return $J(\pi_{Gen}) = \mathbb{E}_{\tau \sim \pi_{Gen}}[R_{Gen}(\mathcal{T})]$.

**Training.** We train the policy $\pi_{Gen}$ using Group Relative Policy Optimization (GRPO). For an input $(Q, S')$, we generate a group of $G$ trajectories $\{\tau_1, \ldots, \tau_G\}$, where each trajectory $tau_i$ consists of a sequence of states and actions $(s_0^i, a_0^i, s_1^i, \ldots)$. The GRPO objective for trajectories is defined as:

$$J_{\text{GRPO}}(\theta) = \mathbb{E}_{\substack{(Q,S') \sim \mathcal{D}, \\ \{\tau_i\}_{i=1}^G \sim \pi_{\theta_{\text{old}}}}} \left[ \frac{1}{G} \sum_{i=1}^G \sum_{t=0}^{|\tau_i|-1} \sum_{j=1}^{|a_t^i|} \min \left( \frac{\pi_\theta(a_{t,j}^i|s_t^i, a_{t,<j}^i)}{\pi_{\theta_{\text{old}}}(a_{t,j}^i|s_t^i, a_{t,<j}^i)} A_i, \text{clip} \left( \frac{\pi_\theta(a_{t,j}^i|s_t^i, a_{t,<j}^i)}{\pi_{\theta_{\text{old}}}(a_{t,j}^i|s_t^i, a_{t,<j}^i)}, 1-\epsilon, 1+\epsilon \right) A_i \right) \right]$$

(4)

where $a_{t,j}^i$ is the $j$-th token of action $a_t^i$ in trajectory $\tau_i$, and $A_i$ is the advantage for the entire trajectory, computed based on the relative rewards of all trajectories within the group. The reward signal $R_{gen}(\tau)$ used to compute $A_i$ is derived solely from execution outcomes, encouraging the agent to prioritize both syntactic validity and semantic correctness:

$$R_{gen}(\tau) = \begin{cases} 1.0 & \text{if final query is valid and execution correct} \\ 0.0 & \text{if valid but incorrect} \\ -1.0 & \text{if invalid} \end{cases}$$

This coarse but decisive feedback gives the agent freedom to discover effective reasoning strategies without being constrained to annotated step-level traces.

**Interactive Reasoning.** The agent is grounded in the ReAct paradigm (Yao et al., 2023), interleaving reasoning and acting in a Think-Act-Observe loop. This iterative structure transforms SQL generation from a one-shot translation into a dialogue with the database, enabling robust recovery from errors.

## 3.3 VALIDATION AGENT: VERIFICATION AND SELECTION

The Validation Agent selects the optimal solution from the multiple candidates generated. Its **input** is the set of $N$ candidate trajectories $\{\tau_1, \ldots, \tau_N\}$ and the original question $Q$. Its **output** is the single best trajectory, $\tau_{\text{final}}$. We employ a Generative Verifier $V$, reframing verification as a next-token prediction task that leverages the base model's own capabilities.

**Training and Inference** The Validation Agent is trained via SFT to generate a single token response: "Yes" for a correct trajectory or "No" for an incorrect one, conditioned on the question and trajectory. The prompt structure is in Appendix C.1.

At inference time, the agent's score for a trajectory $\tau_i$ is the average log probability of the "Yes" token across $M$ stochastic reasoning rounds :

$$V(\tau_i) = \frac{1}{M} \sum_{j=1}^M P(y_j = \text{"Yes"}|\tau_i, Q)$$

(5)

The trajectory with the highest confidence score is selected as the final answer:

$$\tau_{\text{final}} = \underset{i \in \{1,\dots,N\}}{\arg\max} \; V(\tau_i) \qquad (6)$$

This method effectively turns the generative model into a high-quality reranker, capable of discerning the most plausible and accurate reasoning path among many alternatives.

## 4 EXPERIMENT

### 4.1 EXPERIMENT SETUP

**Implementations.** Our experimental setup consists of three distinct agents: a Grounding Agent, a Generation Agent, and a Validation Agent. All models were implemented using PyTorch and trained on NVIDIA H800 GPUs. The Grounding and Generation Agents were trained using Reinforcement Learning (RL). The Grounding Agent was developed with the Verl framework (Sheng et al., 2024), using training data prepared with SQLGlot (Mao, 2023). The Generation Agent utilized a framework adapted from SkyRL (Liu et al., 2025a). The prompt structures for these agents are detailed in Appendix F and Appendix G, with specific training hyperparameters listed in Appendix B.

The Selection Agent was trained via full-parameter Supervised Fine-tuning (SFT) of the Qwen2.5-Coder-7B-Instruct model (Hui et al., 2024). The dataset for this agent was constructed by generating multiple trajectories for each question in the BIRD training set using our trained Generation Agent. Positive and negative examples were then selected based on final execution results. The prompt format for the Validation Agent is shown in Appendix K, and its training hyperparameters are also detailed in Appendix B. For the inference phase, we explicitly configure the sampling parameters to ensure reproducibility. Specifically, we set the number of rollouts for the Generation Agent to $G = 8$. Similarly, the Validation Agent employs $M = 8$ stochastic reasoning rounds for probability estimation. It is worth noting that while scaling $G$ (e.g., to 16 or 32) can yield marginal performance improvements, we adopted $G = 8$ as the standard setting to maintain a balance between accuracy and computational efficiency.

**Benchmark Dataset.** All experiments are conducted on the BIRD (Li et al., 2023),Spider 1.0 (Yu et al., 2019) and Spider-DK (Gan et al., 2021) dataset. We adapt Bird for in-domain evaluation and use Spider, Spider-DK as an out-of-domain dataset. Details on these datasets are in Appendix C.2

**Evaluation Metric.** We evaluate model performance using Execution Accuracy (EX), which is the primary metric for correctness. A predicted SQL query receives a score of 1 if its execution result is identical to the execution result of the ground-truth query, and 0 otherwise. The final score is the percentage of correctly executed queries.

**Baseline models.** To contextualize the performance of our method, **MARS-SQL**, we conduct a comprehensive comparison against a diverse set of models. These are organized into three distinct categories: Base models, High-performing closed-source systems, and Trained open-source models. **Base Models:** This category includes foundational large language models used without task-specific fine-tuning to establish a performance baseline. We evaluate O3-mini, GPT-4o (OpenAI, 2023), GPT-5 and Qwen2.5-coder-7B (Hui et al., 2024). These results help gauge the inherent Text-to-SQL capabilities of modern LLMs before specialized training. **Closed Source Multi agent framework:** This category consists of systems that leverage powerful proprietary models via APIs, representing the upper bound of performance achievable with leading commercial technology. These methods, such as CHESS (Talaei et al., 2024), OpenSearch-SQL (Xie et al., 2025b), XiYan-SQL (Liu et al., 2025b), and CHASE-SQL (Pourreza et al., 2024), typically employ sophisticated frameworks and prompting techniques. This comparison situates our open-source multi-agent framework performance against industry-leading systems. **Open Source Agent Framework:** This group comprises leading open-source models specifically fine-tuned for the Text-to-SQL task, representing the current state-of-the-art in the research community. These models, including CodeS (Li et al., 2024b), Share (Qu et al., 2025), OmniSQL (Li et al., 2025a), Arctic-Text2SQL-R1 (Yao et al., 2025), and Reasoning SQL (Pourreza et al., 2025), employ various advanced training methodologies. Comparing **MARS-SQL** against these systems directly assesses its competitiveness and advancements over existing specialized methods.

Table 2: Main results on the BIRD-dev, Spider-test, and Spider-DK benchmarks. We report Execution Accuracy (%). 'Thinking?' indicates whether the method uses a multi-step reasoning process. Our model is compared against base models and other advanced open and closed-source methods. **Bold** indicates the best result, and underline indicates the second best.

| Model | Params | Thinking? | Training set | Bird-dev (%) | Spider-test (%) | Spider-DK (%) | Sparc(%) |
|---|---|---|---|---|---|---|---|
| *Base Models* | | | | | | | |
| O3-mini | - | Yes | - | 61.34 | 78.82 | 71.77 | 67.0 |
| Qwen-2.5-coder | 7B | No | - | 54.56 | 75.87 | 61.31 | 64.1 |
| GPT-4o | - | No | - | 61.90 | 77.10 | 72.9 | - |
| GPT-5 | - | No | - | 65.45 | 78.39 | 66.92 | 61.8 |
| *Closed-source Multi agentic framework* | | | | | | | |
| CHESS | - | No | - | 65.00 | 87.2 | - | |
| OpenSearch-SQL+ GPT-4o | - | No | - | 69.30 | 87.1 | - | - |
| XiYan-SQL | - | No | - | 73.34 | 89.65 | - | - |
| CHASE-SQL + Gemini | - | Yes | - | 74.90 | 87.6 | - | - |
| *Open Source Agentic Framework* | | | | | | | |
| Qwen-2.5-coder+SFT | 7B | No | Bird | 61.08 | 76.38 | 58.69 | - |
| Qwen-2.5-coder+RL | 7B | Yes | Bird | 62.32 | 77.85 | 66.54 | - |
| CodeS | 7B | No | Spider | 57.17 | 80.3 | 72.0 | - |
| Share | 8B | No | Bird | 64.14 | 85.90 | 75.3 | - |
| OmniSQL | 32B | No | OmniSQL | 64.5 | 87.60 | 76.1 | - |
| Arctic-Text2SQL-R1 | 32B | Yes | Bird+Spider | 70.50 | 88.70 | **80.6** | - |
| Reasoning SQL | 14B | Yes | Bird | 72.29 | 81.43 | 73.03 | - |
| **MARS-SQL** | **21B (3x7B)** | **Yes** | Bird | **77.84** | **89.75** | 78.13 | **85.78** |

## 4.2 MAIN RESULTS

As presented in Table 2, our method, **MARS-SQL**, trained solely on the BIRD training set, achieves state-of-the-art execution accuracy on both the Bird-dev (77.84%) and the Spider-test (89.75%). Additionally, it obtains the second-highest score on the Spider-DK benchmark with 78.13%.

**In-Domain Performance on BIRD-dev.** On the in-domain BIRD-dev set, **MARS-SQL** establishes a new state-of-the-art with an execution accuracy of **77.84%**. This result represents a significant improvement of 5.55% over the next best open-source competitor, Reasoning SQL (72.29%). More impressively, our 7B model also outperforms all listed closed-source solutions, including the strong CHASE-SQL + Gemini (74.90%). This demonstrates the superior effectiveness of our training methodology on this complex, real-world benchmark.

**Out-of-Domain Generalization.** The out-of-domain generalization of **MARS-SQL** is particularly noteworthy, demonstrated by its strong performance on both the Spider-test and Spider-DK benchmarks. On the broad Spider-test set, it achieves a state-of-the-art score of **89.75%**, showcasing exceptional generalization to unseen schemas and question types. This robustness extends to the specialized Spider-DK benchmark—which tests for implicit domain knowledge—where **MARS-SQL** secures a competitive second-highest score of **78.13%**. Crucially, these results were achieved without any exposure to the Spider training set. This contrasts with competitors like Arctic-Text2SQL-R1, which required training on Spider data (from which Spider-DK is derived) to achieve its high scores. Therefore, our model's performance highlights that training solely on the diverse BIRD dataset effectively equips it for broad cross-domain and knowledge-intensive challenges.

## 4.3 ABLATION STUDIES

**Multi-agent frame components analysis.** We conduct a systematic ablation study to validate the contribution of each key component in our **MARS-SQL** framework, with results presented in Table 3. The analysis confirms that both the Grounding agent and the Generative Validation Agent are critical; removing either leads to a significant degradation in performance on all benchmarks. Notably, our purpose-built validation agent substantially outperforms a strong alternative like Self-Consistency (77.84% vs. 72.93% on BIRD-dev), highlighting the benefits of a specialized validation agent. Crucially, the results reveal a powerful synergistic effect, as the final performance gain of the full model is far greater than the sum of the individual components' contributions. This indicates that the Grounder enables the Generator to produce higher-quality trajectories, which our validation agent can then more accurately select. These findings validate our central hypothesis that each agent in the **MARS-SQL** framework is indispensable for achieving state-of-the-art performance.

Table 3: Ablation study on the components of our multi-agent framework. We evaluate the contribution of each agent (Grounder, Verifier) and training strategy (SFT vs. RL). The final row, **MARS-SQL**, represents our full proposed model, demonstrating the synergistic effect of all components.

| Configuration | Model Size | Bird dev (%) | Spider test (%) | Spider DK (%) |
|---|---|---|---|---|
| *Ablating Core Components* | | | | |
| Generator Only (Base) | 7B | 66.37 | 80.11 | 69.91 |
| w/o verifier (Grounding agent+ RL Generator) | 7B | 68.71 | 80.72 | 70.65 |
| w/o Grounder (RL Generator + Verifier) | 7B | 69.75 | 89.19 | 77.01 |
| w/ Self-Consistency (instead of Verifier) | 7B | 72.93 | 83.51 | 73.08 |
| **MARS-SQL(Full Framework)** | **21B (3x7B)** | **77.84** | **89.75** | **78.13** |

**Influence of different max interaction turns.** We then study the impact of the maximum interaction

Figure 3: Execution accuracy on Bird-dev of models fine-tuned with different maximum interaction turns (T), evaluated at inference turn limits of 1, 5, and 10. **'Greedy'** uses a single generation trajectory ($N = 1$) without validation; **'Selected'** denotes the final trajectory chosen by the Validation Agent from $N = 8$ candidates; and **'Best of N'** represents the oracle upper bound where the question is considered correct if any of the $N$ candidates matches.

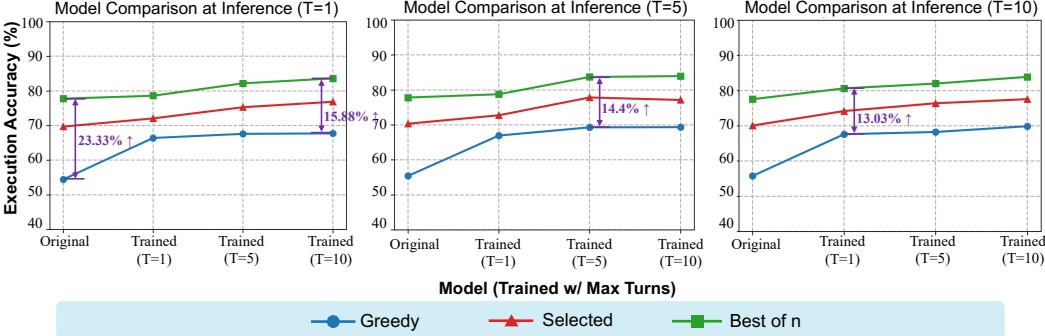

turns (T) during Reinforcement Learning. The results are visualized in Figure 3, with full details provided in Appendix I. As shown, increasing T from 1 to 10 consistently improves both Greedy and Best of 8 accuracy. Notably, our model trained with T=10 significantly outperforms models trained with fewer turns across all inference settings. For instance, at Inference (T=10), it achieves 69.88% Greedy accuracy, surpassing the T=1 model (67.60%) and the base model (55.76%). Furthermore,

this process enhances single-pass reliability by narrowing the gap between Best of 8 (potential) and Greedy (actual) performance. This gap shrinks from a substantial 23.33% in the base model to 12.19% in the T=1 model at Inference (T=1). Training with a larger T reinforces this effect, making the model's greedy output more aligned with its optimal potential, thereby improving its dependability.

Figure 4: Comparison of different selection strategy.

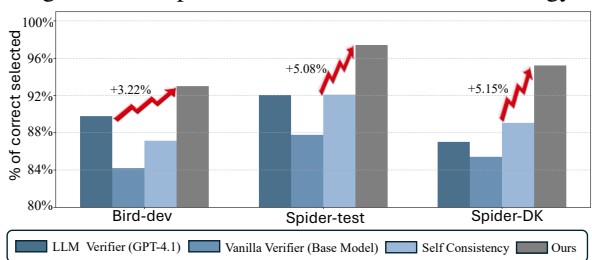

**Selection methods analysis.** To validate the effectiveness of our Generative Validation Agent, we compare it against several alternative selection strategies, as illustrated in Figure 4. While common approaches such as Self-Consistency or using a powerful LLM as a Judge (e.g., GPT-4.1) provide a reasonable baseline, their performance is both suboptimal and inconsistent across the different benchmarks. In stark contrast, our fine-tuned Generative Validation Agent consistently outperforms all other methods by a significant margin. On the challenging Spider-test, it achieves a correct selection rate of 97.15%, a substantial improvement over the next-best strategy's 92.09%.

Similar significant gains are observed on both BIRD-dev and Spider-DK. This consistent superiority demonstrates the stability and robustness of our specialized approach. Unlike general-purpose models or heuristic-based methods, our validation agent reliably identifies the most accurate reasoning trajectory, making it a critical component for achieving state-of-the-art performance. Full execution accuracy results for each method are detailed in Appendix M.

## 5 RELATED WORK

**LLMs for Text-to-SQL** The rise of Large Language Models (LLMs) has brought notable progress to Text-to-SQL tasks, moving past traditional sequence-to-sequence approaches. Recent studies emphasize in-context learning, where strategies such as Chain-of-Thought (CoT) prompting are used to break down complex queries into intermediate reasoning steps (Tai et al., 2023; Dong et al., 2023). Frameworks like DIN-SQL (Pourreza & Rafiei, 2023) and DAIL-SQL (Gao et al., 2023) have systematically explored prompt engineering and multi-stage pipelines that include schema linking, generation, and refinement to boost performance. Building on these ideas, more recent studies (Wang et al., 2025a; Deng et al., 2025; Gao et al., 2025; Xie et al., 2025b) move toward structured, multi-step workflows that better match the complexity of real databases and diverse queries. Our work adopts this decompositional philosophy but shifts away from static prompting by introducing a dynamic, learning-based agentic system.

**Multi-Agent systems** Large Language Models (LLMs) have enabled sophisticated multi-agent systems by adopting specialized roles via in-context prompting (Wang et al., 2024; Min et al., 2022). Our focus is on goal-oriented problem-solving frameworks, rather than social simulations (Zhang et al., 2024; Hua et al., 2024), where tasks are divided among collaborating agents. The complexity of these collaborations has grown from simple debating (Du et al., 2023) to structured workflows with the use of tools, such as software development agents ChatDev (Qian et al., 2024), MetaGPT (Hong et al., 2024) and CollabUIAgent (He et al., 2025). Other notable approaches include the generic framework AutoGen (Wu et al., 2023) and the dynamic cooperation in AutoAgents (Chen et al., 2024). Following this established paradigm, we propose a specialized pipeline for Text-to-SQL using Grounder, Generator, and Verifier agents.

**Reinforcement Learning** Reinforcement Learning (RL) is increasingly used to enhance the complex reasoning capabilities of LLMs, especially when combined with chain-of-thought prompting (Wei et al., 2023; OpenAI, 2024). This approach has proven highly effective, achieving state-of-the-art results in fields like mathematics and code generation (Qin et al., 2023; Zhao et al., 2024). Typical approaches fine-tune models with policy gradient methods such as PPO or GRPO, rewarding logical soundness or correct outcomes (Shao et al., 2024; DeepSeek-AI et al., 2025). While PPO is a common choice, GRPO offers advantages by being less prone to high variance and more memory-efficient, as it does not require loading an additional critic model. In parallel, interactive reasoning paradigms like ReAct (Yao et al., 2022) leverage prompting-based Think–Act–Observe loops to enable tool use and self-correction, but without explicit policy training. While Text-to-SQL requires similarly complex reasoning, explicit RL for this domain remains underexplored. Our work addresses this gap by training the Generator agent's policy with execution-based rewards, enabling robust, stateful query generation and dynamic self-correction.

**Test-Time Scaling** To enhance performance without the cost of retraining, many researchers have focused on inference-time techniques. Self-consistency, for instance, has become a popular method where multiple reasoning paths are sampled and the final answer is chosen by majority vote (Wang et al., 2023). This concept has been further refined by verification and reranking methods, which employ an external mechanism or model to score and select the best candidate from a pool of outputs (Zheng et al., 2023; Gu et al., 2025). Our approach builds on the recent innovation of Generative Verifiers (Zhang et al., 2025b). Instead of a voting process or a separate classifier, our Validation Agent reframes selection as a next-token prediction problem. It assesses each potential solution trajectory by calculating the probability of the model generating a "Yes" token, ultimately selecting the trajectory with the highest confidence score.

## 6 CONCLUSION

In this work, we present MARS-SQL, a multi-agent framework that addresses the limitations of static, single-pass Text-to-SQL methods. By decomposing the task into schema grounding, interactive query generation, and final verification, our framework achieves robust performance through specialized agents. The core of our system is the Generator agent, which uniquely leverages reinforcement learning within a ReAct-style Think–Act–Observe loop to enable dynamic reasoning and self-correction. **MARS-SQL** established new state-of-the-art execution accuracies on BIRD (77.84%) and Spider (89.75%), demonstrating strong cross-domain generalization by achieving its Spider result without any training on the Spider dataset. Ablation studies further demonstrate that each agent plays a distinct role, and their combination delivers substantial gains over any single component. These findings highlight the promise of moving from static, one-shot generation toward interactive, multi-agent problem solving as a foundation for building more reliable data-centric AI systems.

## REPRODUCIBILITY STATEMENT

To ensure the reproducibility of our work, we are committed to making our code and models publicly available upon publication. All experiments were conducted on publicly accessible and widely used benchmarks: BIRD (Li et al., 2023), Spider (Yu et al., 2019), and Spider-DK (Gan et al., 2021). The primary evaluation metric is Execution Accuracy (EX), a standard in the Text-to-SQL field. Key details regarding our implementation, including the multi-agent framework architecture, prompt structures for each agent, and training hyperparameters, are described in the main body of the paper and further detailed in the Appendix. Our methodology, including the use of Group Relative Policy Optimization (GRPO) and the specific design of our reward functions, is explicitly formulated to facilitate replication by future research.

## ETHICS STATEMENT

The primary goal of this research is to develop more robust and reliable Text-to-SQL systems, aiming to democratize data access for non-expert users and reduce barriers to data-driven insights. Our work relies exclusively on publicly available datasets (BIRD and Spider) that are standard academic benchmarks and do not contain personally identifiable information or sensitive user data. We acknowledge that any Text-to-SQL system, including ours, carries an inherent risk of generating incorrect or unintended queries, which could lead to flawed analysis if deployed without human oversight. However, our framework's emphasis on dynamic self-correction and robust verification is a direct attempt to mitigate these risks and improve the reliability of AI agents interacting with databases. We believe the potential benefits of making complex data more accessible outweigh the risks, and we encourage the deployment of such systems in a manner that includes human-in-the-loop validation for critical applications.

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

# A   THE USE OF LARGE LANGUAGE MODELS

Large Language Models (LLMs) were utilized in a limited, assistive capacity for specific tasks in this project. For manuscript preparation, the authors supplied their own draft to an LLM, which then provided suggestions to improve grammar, enhance clarity, and ensure an academic tone. The LLM was also used to generate a list of potential titles for inspiration, though the final title was conceived and refined by the authors and not taken directly from any single output. In the implementation phase, an LLM served as a coding assistant by offering code completions and debugging support. However, all final code, experimental design, and validation were implemented and verified exclusively by the authors. It is important to emphasize that LLMs were **NOT** used for core scientific contributions, such as generating research ideas, designing experiments, or conducting the literature review. All conceptual work and experimental design originated solely with the authors.

# B   TRAINING DETAILS

This section provides the detailed hyperparameters used for training our three agents. All agents were trained on NVIDIA H800 GPUs.

## B.1   GROUNDING AGENT

The Grounding Agent was trained using Reinforcement Learning baesd on Qwen2.5-Coder-7B-Instruct. Its primary role is to identify the correct database schema entities relevant to the user's question. The training was conducted using the Verl framework (Sheng et al., 2024). The hyperparameters for the RL training and data generation phases are detailed in Table 4.

**Training Steps and Convergence:** We trained the Grounding Agent for **600 steps** with a batch size of 64. During training, we observed clear stability and convergence patterns in the reward curves; the reward consistently increased and then plateaued, indicating that the policy was successfully optimized.

## B.2   GENERATION AGENT

The Generation Agent was also trained using Reinforcement Learning based on Qwen2.5-Coder-7B-Instruct, leveraging a training framework adapted from SkyRL (Liu et al., 2025a). This agent is responsible for generating the SQL query trajectories. Its training and data generation hyperparameters are identical to those of the Grounding Agent, as shown in Table 4.

**Training Steps and Convergence:** This agent was trained for **160 steps** with a batch size of 64. Similar to the Grounding Agent, the reward curve demonstrated stable convergence within this efficient training phase.

Table 4: Hyperparameters for Grounding and SQL Agent RL Training.

| Parameter | Value |
|---|---|
| *Training Parameters* | |
| Learning Rate | $1 \times 10^{-6}$ |
| Batch Size | 128 |
| *Trajectory Rollout Parameters* | |
| Temperature | 0.6 |
| Top-p | 0.95 |

## B.3   VALIDATION AGENT

The Validation Agent was trained via Supervised Fine-tuning (SFT) to select the best SQL query from the candidates generated by the SQL Agent. We performed a full-parameter fine-tuning of the `Qwen2.5-Coder-7B-Instruct` model (Hui et al., 2024) using the Llama Factory framework.

The SFT training hyperparameters are listed in Table 5, and the parameters for generating its training dataset are in Table 6.

Table 5: Hyperparameters for Verify Agent SFT.

| Parameter | Value |
|---|---|
| Base Model | `Qwen2.5-Coder-7B-Instruct` |
| Epochs | 3 |
| Learning Rate Scheduler | Cosine |
| Initial Learning Rate | $1 \times 10^{-5}$ |
| Effective Batch Size | 4 |
| *Per-device Batch Size* | *1* |
| *Gradient Accumulation* | *2 steps* |
| Precision | `bf16` |
| Optimization | DeepSpeed ZeRO Stage 3 |

Table 6: Hyperparameters for Verify Agent Dataset Generation.

| Parameter | Value |
|---|---|
| Candidates per Question | 16 |
| Temperature | 0.7 |
| Top-p | 0.9 |
| Top-k | 50 |

## C    DATASET

### C.1    TRAINING DATASET

Our training data is derived from the Bird benchmark, which comprises 9,428 question-SQL pairs. To ensure high quality, we first filtered this dataset by removing samples flagged as incorrect (Pourreza et al., 2025; Li et al., 2024b) by both Gemini-2.5-pro and GPT-4o, resulting in a clean set of 8,036 training examples. From this set, we constructed the fine-tuning data for the grounding task. For each of the 8,036 question-database pairs, we generated a distinct training instance for every table within that database. This process resulted in a large-scale dataset of 90,102 individual data points. For each point, the ground truth—whether a table is relevant and which of its columns are used—was programmatically extracted from the gold SQL query using the SQLGlot parser.

We constructed a specialized dataset for training the Verifier via Supervised Fine-Tuning (SFT). First, for each question in our filtered BIRD training set, we used both our fine-tuned Generator agent and the initial base model to perform inference, generating a diverse pool of 16 candidate trajectories per question. This ensures the Verifier is exposed to a wide range of reasoning paths, both correct and flawed. From this pool, we curated a preference dataset by selecting one positive example (a trajectory leading to a correct execution result) and one negative example (a trajectory leading to an incorrect result) for each question. We mix the order of correct and incorrect trajectories in each pair at random to prevent order bias during training. Since the number of cases containing both correct and incorrect trajectories is limited, some questions yield only flawed trajectories. In such cases, we add the ground truth SQL query in the prompt as a suggestion to help the model generate proper trajectories. We applied best-of-N and worst-of-N (Gui et al., 2024) strategies to select both positive and negative examples. This process yielded a final dataset of approximately 16,000 training instances. Each instance is a triplet containing the user's question, the full interaction trajectory (including all [Think], [SQL], and [Observation] steps), and the final execution result.

### C.2    EVALUATION DATASET

BIRD is a large-scale, realistic benchmark designed to evaluate modern Text-to-SQL systems. It features complex databases (33.4 GB across 95 databases), questions from 37 professional domains,

and imperfect real-world data values requiring robust handling. BIRD uniquely emphasizes the generation of both correct and efficient SQL queries, making it an ideal testbed for our framework. Our primary evaluations are performed on its development set, which contains 1,534 examples.

Spider 1.0 is a comprehensive, cross-domain benchmark containing 10,181 questions and 5,693 unique complex SQL queries across 200 multi-table databases. It serves as a standard for evaluating cross-domain Text-to-SQL performance. For our evaluation, we use the official test set, which includes 2,147 examples.

Spider-DK, an extension of Spider, is designed specifically to test a model's ability to handle queries requiring implicit domain knowledge. It comprises samples from the Spider development set that were manually modified to depend on real-world information for correct interpretation. This benchmark simulates scenarios where user queries rely on specific domain context. We evaluate our model on the Spider-DK test set, which contains 535 examples.

# D TRAINING EFFICIENCY AND RESOURCE ANALYSIS

To address concerns regarding the computational resources required for our multi-agent framework, we provide a detailed breakdown of the training time and a comparative analysis of data efficiency. All experiments were conducted on a node equipped with $4 \times$ NVIDIA H800 GPUs.

## D.1 COMPUTATIONAL COST BREAKDOWN

Contrary to the perception that training multiple agents is prohibitively resource-intensive, our framework is designed for rapid convergence. As detailed in Table 7, the entire specialized training pipeline—including the SFT for the Validation Agent and GRPO-based Reinforcement Learning for both the Grounding and Generation Agents—completes in approximately **13 hours**. This represents a modest one-time computational cost, especially considering the significant performance gains achieved.

Table 7: One-Time Training Cost breakdown on $4 \times$ NVIDIA H800 GPUs.

| Agent | Method | Training Steps | Batch Size | Est. Training Time |
|---|---|---|---|---|
| Validation Agent | SFT | ∼10k | 4 | 1 h |
| Grounding Agent | GRPO | 600 | 64 | 4 h |
| Generation Agent | GRPO | 160 | 64 | 8 h |
| **Total** | | | | **∼13 h** |

## D.2 DATA EFFICIENCY AND COMPARATIVE ANALYSIS

The efficiency of MARS-SQL stems from its ability to learn diverse reasoning and self-correction behaviors through interaction and self-play, rather than relying on massive-scale supervised datasets.

Table 8 compares our framework against standard single-agent SFT approaches. While standard SFT on the BIRD training set (12k examples) takes only 2 hours, it yields a significantly lower execution accuracy (EX) of 61.08%. Scaling up SFT, as seen in methods like OminiSQL (utilizing 2.5M examples), requires approximately 20 days of training yet only reaches 64.50% EX.

In contrast, MARS-SQL achieves a state-of-the-art EX of **77.84%** using only **35k** LLM-labeled examples and 13 hours of training. To match this performance level using a single-agent SFT-only paradigm, we conservatively estimate—based on scaling laws—that it would require approximately **15 million** synthetic examples and **3–4 months** of training time on the same hardware. Thus, our multi-agent RL framework offers orders of magnitude better data and compute efficiency.

# E INFERENCE EFFICIENCY AND PRACTICALITY ANALYSIS

In this section, we provide a comprehensive analysis of the efficiency and practicality of the MARS-SQL framework. We focus on the cost-benefit trade-offs and demonstrate that the proposed multi-

Table 8: Cost and efficiency analysis compared with single-agent SFT baselines on Bird-Dev.

| Method | Annotation Source | Data Size | Training Time (wall) | Dev EX (%) |
|---|---|---|---|---|
| Original (Baseline) | — | — | — | 54.56 |
| SFT on BIRD-train | Human | 12,000 | $\sim$2 h | 61.08 |
| Large SFT (e.g., OminiSQL) | LLM + Human | 2,500,000 | $\sim$20 days | 64.50 |
| **MARS-SQL (Ours)** | LLM | **35,000** | $\sim$**13 h** | **77.84** |

agent system provides a flexible and effective solution compared to counterpart methods. Our analysis covers three key aspects: (1) performance comparison under a normalized time budget, (2) adjustable cost–accuracy trade-offs, and (3) potential system-level optimizations.

### E.1 BASELINE TIME AND TOKEN COST ANALYSIS

We first present the latency breakdown for our standard SOTA-performing configuration ($N_g = 8$ trajectories, $N_v = 8$ validation samples). As shown in Table 9, the average end-to-end latency is 22.12 seconds per query to achieve 77.84% accuracy.

Table 9: Average End-to-End Latency Analysis of MARS-SQL on the BIRD dev set (Hardware: 1x A6000, num_cpus=32). Times represent the average latency to generate one SQL query.

| Stage | Avg. Time (s) | Description |
|---|---|---|
| 1. Grounding Agent | 0.78s | 1 call per query |
| 2. Generation Agent | 18.77s | Generating $N_g = 8$ trajectories |
| 3. Validation Agent | 2.58s | Validating $N_g = 8$ trajectories ($N_v = 8$ samples each) |
| *Ref: SQL Exec Time* | *(2.37s)* | *Avg. time to execute the ground truth SQL* |
| **Total (End-to-End)** | **22.12s** | **Sum of all stages** |

The token consumption is analyzed in Table 10. The Generation Agent, utilizing a multi-turn "Think-Act-Observe" loop, accounts for the majority of the token usage.

Table 10: Average Token Cost Analysis per Query.

| Stage | Avg. Tokens | Description |
|---|---|---|
| 1. Grounding Agent | 875 | Prompt + Schema + Question + Output |
| 2. Generation Agent | 9,200 | $N_g = 8\times$ (Prompt + Schema + Question + Traj.) |
| 3. Validation Agent | 3,250 | $N_g = 8 \times N_v = 8\times$ (Prompt + Trajectory) |
| **Total (Avg.)** | **13,325** | **Sum of all components** |

### E.2 PERFORMANCE COMPARISON UNDER NORMALIZED TIME BUDGET

To verify the effectiveness of our multi-agent design, we benchmark MARS-SQL against both a supervised fine-tuning (SFT) model and a closed-source model under an equal time budget ($\approx$ 22s).

- **MARS-SQL:** Uses the standard setting ($N_g = 8$, $N_v = 8$).
- **SFT Model (`Qwen-SFT`):** Uses the 22s budget to generate 16 independent samples and selects the most self-consistent one.
- **Closed-Source Model (`GPT-5`):** Uses the 22s budget to make 4 API calls and selects the most self-consistent one.

As shown in Table 11, baselines fail to match the performance of MARS-SQL even when granted an equivalent time budget. This indicates that the superior accuracy of MARS-SQL (77.84%) stems from its structured multi-agent reasoning workflow rather than merely increased inference time.

Table 11: Accuracy Comparison with Normalized Time Budget ($\approx 22s$).

| Method | Configuration | Avg. Latency | Exe. Acc. (%) |
|---|---|---|---|
| `Qwen-SFT` (Self-Consistency) | SFT + 16 Samples | $\approx 22.0$s | 64.2% |
| `GPT-5` (Self-Consistency) | 4 API calls | $\approx 22.0$s | 69.3% |
| **MARS-SQL (Ours)** | **Multi-Agent RL** | **22.12s** | **77.84%** |

### E.3 ADJUSTABLE COST–ACCURACY TRADE-OFFS

The latency reported in Table 9 represents a performance-oriented configuration. MARS-SQL allows for flexible deployment by adjusting the number of generation trajectories ($N_g$) and validation samples ($N_v$). Table 12 illustrates these trade-offs.

Table 12: Tunable Cost-Accuracy Curve for MARS-SQL.

| Mode | Params ($N_g, N_v$) | Latency | Acc. (%) | Characteristic |
|---|---|---|---|---|
| Fast | (1, 1) | 3.1s | 68.71% | High speed, outperforms SFT |
| Balanced | (4, 4) | 11.5s | 74.90% | Balanced cost-benefit |
| **SOTA (Ours)** | **(8, 8)** | **22.1s** | **77.84%** | **Maximum accuracy** |
| Over-Sampling | (16, 8) | 42.8s | 77.84% | Diminishing returns |

Users can select a "Fast" setting to achieve a $\approx$3-second response that still surpasses the greedy SFT baseline, or invest more computational resources for maximum performance.

### E.4 SYSTEM-LEVEL OPTIMIZATION

The latency metrics presented above assume a sequential, single-query execution, serving as a conservative upper bound. In practical multi-user deployments, **MARS-SQL** can achieve higher throughput through system-level optimizations:

1. **Pipeline Parallelism:** The Grounder, Generator, and Validator agents can process different queries in parallel, creating a pipeline for incoming requests.

2. **Batched Validation:** The $N_g \times N_v$ validation calls are embarrassingly parallel and can be fused into batched requests to reduce amortized costs.

Table 13: Sequential vs. System-Optimized Deployment (Conceptual Comparison).

| Deployment | Execution Pattern | Est. Latency | Est. Throughput |
|---|---|---|---|
| **Sequential** (No optimization) | Grounder $\rightarrow$ Generator $\rightarrow$ Verifier (End-to-End) | $\approx 22.1$s | $\approx 2.7$ queries/min |
| **System-Optimized** (Pipeline + Batch) | Pipelined stages; Batched validation | $\approx 12$–15s | $\approx 4$–5 queries/min |

Table 13 estimates that with these optimizations, the effective per-query latency can be reduced by approximately 40–60%, significantly improving throughput on a single GPU node.

## F TABLE LEVEL GROUNDING

Table 14 details the prompt for our RL-trained Schema Grounding Agent, which elicit a step-by-step reasoning process during inference. It instructs the agent to analyze a given table's schema in the context of the user's question and any external knowledge. The agent is required to first articulate its analysis within '<think>' tags, followed by a final, parsable decision in '<answer>' tags. This output must specify the table's relevance ('Y'/'N') and, if applicable, a Python list of useful columns.

---

**Prompt for Table-level Schema Linking**

**User:**

You are doing table level schema linking. Given a table with schema information and the task, you should think step by step and decide whether this table is related to the task.

Your thought process should be enclosed in `<think></think>` tags, and your final decision in `<answer></answer>` tags.

For the answer, first state 'Y' for relevant or 'N' for not relevant. If relevant, also provide a Python list of the column names you believe are most useful.

Example of a final answer format:
```
<answer>
Y
["player_name", "team_name", "matches_played"]
</answer>
```

or

```
<answer>
N
</answer>
```

Here is the information for the current task:

**### Table Information:**
*{table_info}*
**### User Question:**
*{task}*
**### External Knowledge (if any):**
*{external}*

---

**Assistant:**

Let me solve this step by step.
`<think>`

---

Table 14: The prompt used to guide the agent in the table-level schema linking task. It includes the role description, task instructions, output format examples, and the prefix for the agent's response.

This structured format ensures a transparent and predictable output format crucial for our framework.

Table 15 presents recall and precision statistics for our schema grounding agent, comparing our RL-based approach against the base model and a version trained with Supervised Fine-Tuning (SFT). The results clearly demonstrate the superiority of our method, which achieves exceptionally high recall and precision across all benchmarks. On the complex in-domain BIRD-dev set, our primary concern is recall. Our agent achieves a recall of 97.78%, with only 48 examples failing to identify all required schema components, which we consider a highly effective result. Simultaneously, it maintains a high precision of 90.74%, indicating that the selections are not only comprehensive but also accurate. This strong performance extends to the out-of-domain Spider-test and Spider-DK benchmarks, underscoring the robustness of our RL-trained grounding agent.

## G MULTI-TURN GENERATION

**Evolution of Interaction Turns:** To understand the impact of RL training on the agent's reasoning efficiency, we analyzed the evolution of rollout lengths during the training process. In our setting, each "Think–Act–Observe" cycle corresponds to one database interaction turn, making the average number of interaction turns a proxy for rollout length.

Table 15: Recall and precision statistics after grounding for Bird-dev, Spider-test and Spider-DK. **Recall** measures the percentage of instances where all required columns were identified. **Precision** measures the ratio of required columns to all selected columns, indicating the selection's accuracy.

| Grounding Model | Bird dev | | Spider test | | Spider DK | |
|---|---|---|---|---|---|---|
| | Recall (%) | Precision (%) | Recall (%) | Precision (%) | Recall (%) | Precision (%) |
| Qwen 7B (Base) | 68.59 | 53.45 | 87.48 | 69.22 | 84.25 | 66.54 |
| Qwen 7B + SFT | 74.97 | 67.01 | 90.39 | 78.16 | 88.60 | 72.71 |
| **Qwen 7B + RL (Ours)** | 97.78 | 90.74 | 98.97 | 93.62 | 98.13 | 91.59 |

**Grounding Agent (Single-turn):** The rollout length (token count) exhibited a mild U-shaped pattern. Initially, the output became more concise, followed by a slight lengthening to include only essential schema information. This reflects a refinement of the policy towards precise schema selection rather than reasoning from scratch.

**Generation Agent (Multi-turn):** A distinct trend was observed where the average number of interaction turns consistently decreased and stabilized at a lower level. This indicates that the agent learned to solve problems more directly and recognized when to terminate the search efficiently. This efficiency gain is quantitatively supported by the evaluation on the BIRD-dev set (with a maximum of 5 turns), as shown in Table 16. The RL-trained agent significantly reduces the average turns across all difficulty levels compared to the base model.

Table 16: Comparison of Average Interaction Turns on BIRD-dev (Max Turns = 5) before and after RL training.

| Model | Avg. Turns (Challenging) | Avg. Turns (Moderate) | Avg. Turns (Simple) |
|---|---|---|---|
| Before RL (Base Model) | 2.90 | 2.67 | 2.27 |
| **After RL (Generation Agent)** | **1.82** | **1.71** | **1.45** |

Furthermore, analyzing the specific distribution of turns reveals that the agent learns an adaptive and non-wasteful strategy. As presented in Table 17, while the agent retains the capacity to use multiple turns for complex reasoning, it solves the vast majority of problems (1,116 cases) in a single interaction. Crucially, for the "long-tail" of more difficult queries, the agent robustly applies deeper reasoning, utilizing up to 5 or more turns to arrive at the correct solution. This distribution confirms that the agent is not bound by arbitrary limits but instead dynamically decides the necessary reasoning depth for each specific query.

Table 17: Distribution of Interaction Turns Used by the Generation Agent on BIRD-dev.

| # of Interaction Turns | # of Examples |
|---|---|
| 1 | 1,116 |
| 2 | 174 |
| 3 | 105 |
| 4 | 88 |
| 5+ | 51 |

Table 18 details the comprehensive prompt structure used to guide the agent's multi-turn generation process. The prompt establishes the agent's persona as a data science expert and provides all necessary context, including the database schema, external knowledge, and the user's question. It strictly enforces an output format that requires the agent to vocalize its reasoning within <think> blocks before executing a query in a <sql> block. The database returns feedback in an <observation> block, which the agent uses for subsequent reasoning turns, ultimately providing the final answer in a <solution> block. This iterative structure is designed to facilitate a dynamic, step-by-step problem-solving process.

Figure 5 provides a concrete example of the agent's interactive and self-correcting workflow. The agent initially generates a query with a typographical error in a table name 'fprm'. Upon receiving an 'OperationalError' from the database , it correctly identifies the mistake in its next thought process,

corrects the table name to 'frpm', and re-executes the query. The second attempt returns an empty result , prompting the agent to re-evaluate its filtering logic. It deduces that it incorrectly used a district name as a county name and refines the 'WHERE' clause accordingly. This final attempt successfully retrieves the correct data, leading the agent to output the validated query as its final solution.

**Q: Please list the zip code of all the charter schools in Fresno County Office of Education.**

Figure 5: Example of interactive process in the generation

# H QUALITATIVE ANALYSIS: REASONING BEYOND VALID EXECUTION

A core feature of our Generation Agent is its learned policy to evaluate the *semantic content* and *plausibility* of execution results, rather than merely checking for syntax errors. This allows the agent to continue reasoning even when the database returns a non-empty, non-error result. Below, we present two real-world case studies from the BIRD-dev set illustrating this capability.

## H.1 SCENARIO 1: COMPLEX PROBLEM DECOMPOSITION

In complex queries, a successful execution often represents only an intermediate step in a larger plan. The agent effectively decomposes the problem, using the observation from the first step to inform the subsequent logic.

**Question:** Consider the average difference between K-12 enrollment and 15-17 enrollment of schools that are locally funded, list the names and DOC type of schools which has a difference above this average.

**Turn 1 (Think):** To solve this problem, I need to follow these steps: 1) Calculate the average difference between K-12 enrollment and 15-17 enrollment for all schools. 2) Filter the schools based on the funding type. 3) Calculate the difference for filtered schools. 4) Identify schools above the average.

**Turn 1 (SQL):**

```sql
SELECT AVG("Enrollment (K-12)" - "Enrollment (Ages 5-17)") FROM frpm;
```

**Turn 1 (Observation):** 16.261833

**Turn 2 (Think):** *(The agent recognizes this is an intermediate value, not the final answer.)* The average difference is approximately 16.26. Now, I will filter the schools based on the funding type (locally funded) and calculate the difference for these schools against this average.

**Turn 2 (SQL):**

```sql
SELECT T2.School, T2.DOC FROM frpm AS T1 JOIN schools AS T2 ON T1.CDSCode
    = T2.CDSCode WHERE T2.FundingType = 'Locally funded' AND (T1."
    Enrollment (K-12)" - T1."Enrollment (Ages 5-17)") > 14.92;
```

**Turn 2 (Observation):** [('Circle of Independent Learning', '54'), ('Learning Community Charter', '00')]

**Turn 3 (Think):** The query executed successfully and returned non-empty results. I will now combine the logic into a single nested query for the final answer.

**Turn 3 (Final Solution):**

```sql
SELECT T2.School, T2.DOC FROM frpm AS T1 INNER JOIN schools AS T2 ON T1.
    CDSCode = T2.CDSCode WHERE T2.FundingType = 'Locally funded' AND (T1.
    "Enrollment (K-12)" - T1."Enrollment (Ages 5-17)") > (SELECT AVG(T3."
    Enrollment (K-12)" - T3."Enrollment (Ages 5-17)") FROM frpm AS T3
    INNER JOIN schools AS T4 ON T3.CDSCode = T4.CDSCode WHERE T4.
    FundingType = 'Locally funded')
```

## H.2 SCENARIO 2: DATA PLAUSIBILITY AND SANITY CHECKING

The agent also learns to perform "sanity checks." If a syntactically correct query returns implausible data (e.g., placeholder values), the agent triggers a self-correction loop to verify and refine the query before finalizing the answer.

**Question:** Which cities have the top 5 lowest enrollment number for students in grades 1 through 12?

**Turn 1 (Think):** I need to find the cities with the top 5 lowest enrollment numbers. I will join the tables and order by enrollment ascending.

**Turn 1 (SQL):**

```sql
SELECT T2.City, T1."Enrollment (K-12)" FROM frpm AS T1 INNER JOIN schools
    AS T2 ON T1.CDSCode = T2.CDSCode ORDER BY T1."Enrollment (K-12)" ASC
    LIMIT 5;
```

**Turn 1 (Observation):** [('Markleeville', 1.0), ('Oakland', 1.0), ('Palermo', 1.0)]

**Turn 2 (Think):** *(The agent detects a potential data anomaly.)* The result shows enrollment numbers are very low (1.0), which might indicate an error in the data or the need to filter out non-enrollment entries. Let's check the data to ensure accuracy.

**Turn 2 (SQL):**

```sql
SELECT CDSCode FROM frpm WHERE "Enrollment (K-12)" != 1.0 LIMIT 10;
```

**Turn 2 (Observation):** [('01100170130419'), ('011000170112607')...]

**Turn 3 (Think):** The check confirms that there are legitimate records with other values, and '1.0' likely represents missing or placeholder data. I need to filter these out to get accurate city names.

**Turn 3 (SQL):**

```
SELECT T2.City FROM frpm AS T1 INNER JOIN schools AS T2 ON T1.CDSCode =
    T2.CDSCode WHERE T1."Enrollment␣(K-12)" != 1.0 ORDER BY T1."
    Enrollment␣(K-12)" ASC LIMIT 5;
```

**Turn 3 (Observation):** [('Bishop'), ('Kingsburg'), ('Oakland'), ('Williams'), ('Compton')]

**Turn 4 (Think):** Now, the cities with the lowest enrollment numbers are identified and the data looks plausible. I can provide the final SQL query.

**Turn 4 (Final Solution):**

```
SELECT T2.City FROM frpm AS T1 INNER JOIN schools AS T2 ON T1.CDSCode =
    T2.CDSCode WHERE T1."Enrollment␣(K-12)" != 1.0 ORDER BY T1."
    Enrollment␣(K-12)" ASC LIMIT 5;
```

## I    INTERACTION TURNS IMPACT

Table 19 presents an ablation study on the impact of the maximum number of interaction turns (T) allowed during training and inference. The study compares the performance of the base model against versions fine-tuned with maximum turn limits of T=1, T=5, and T=10. For each training configuration, we report three execution accuracy metrics on the BIRD-dev benchmark: Greedy (a single output), Best of 8 (the highest score among 8 candidates), and Selected (the score after reranking). The results indicate that increasing the number of turns during fine-tuning consistently improves performance. For example, the model trained with T=10 achieves the highest Greedy (69.88%) and Best of 8 (83.89%) accuracy when inferring with 10 turns, demonstrating the value of a larger interaction budget for complex reasoning.

## J    IMPACT OF MULTIPLE CANDIDATE GENERATIONS

To evaluate the impact of generating multiple candidate trajectories, we conduct a "Best-of-N" analysis, where N is the number of parallel rollouts. As shown in Table 20, increasing the number of candidates provides a substantial performance boost. This demonstrates that the exploratory nature of our Generator agent is effective at covering the solution space, with the upper-bound performance (Pass@N) increasing consistently with more samples. The final accuracy, after applying our Generative Validation Agent, also benefits from a larger pool of high-quality candidates to select from.

## K    VALIDATION AGENT

Our Generative Validation Agent is guided by the prompt detailed in Table 21. The prompt instructs the agent to act as an expert SQL data analyst, with the objective of evaluating the logical correctness of a proposed SQL solution for a given problem. Unlike our previous approach, this prompt no longer constrains the agent to reason about a sampled or truncated database. Instead, it assumes the agent evaluates the query's validity against the full database schema and context. The prompt structure provides the agent with the user's question, the candidate SQL solution, and a dedicated field for any relevant "External Knowledge" that might be necessary for a correct evaluation. The output format remains strict, requiring the agent to begin its response with a definitive "Yes" or "No" before any subsequent reasoning.

## L    LLM AS A JUDGE PROMPT

The prompt in Table 22 is used for our baseline selection method LLM as a judge. This prompt is designed to guide the model in identifying the optimal SQL query from a set of generated candidates.

The agent is explicitly instructed to consider each candidate's associated reasoning, the SQL query itself, and most crucially, its execution observation on the database. This emphasis on execution results is paramount, as it allows the agent to distinguish between syntactically correct queries and those that truly provide the correct and complete answer to the user's question, even if a query might appear correct but yields erroneous or empty results. After presenting the user's question and the formatted candidate solutions (each including reasoning, SQL, and execution output), the prompt concludes with strict instructions for the agent to output only the index number of the single best candidate. In cases of ties, the candidate with the lowest index is to be chosen, ensuring a deterministic selection process.

## M  SELECTION METHOD COMPARISON

We compare our proposed Generative Verifier against several strong baselines for trajectory selection, with the results detailed in Table 23. The initial Pass@8 accuracy of our Generator agent's output establishes the theoretical upper bound for any selection method, as it represents the percentage of questions for which at least one of the eight generated trajectories is correct.

---

**Prompt Format for SQL Reasoning**

**Prompt Description:**
You are a data science expert. Below, you are provided with a database schema and a natural language question. Your task is to understand the schema and generate a valid SQL query to answer the question within limited turns. You should breakdown the problem, draft your reasoning process, and generate the solution.
**Database Engine:**
SQLite
**Database Schema:**
{db_details}
This schema describes the database's structure, including tables, columns, primary keys, foreign keys, and any relevant relationships or constraints.
**External Knowledge:**
{external_knowledge}
**Question:**
{question}
**Important Instructions:**
- Make sure you only output the information that is asked in the question. If the question asks for a specific column, make sure to only include that column in the SELECT clause, nothing more.
- The generated query should return all of the information asked in the question without any missing or extra information.
- Before generating the final SQL query, please think how to write the query. It should include detailed considerations such as analysing questions, summarizing relevant findings, brainstorming new ideas, verifying the accuracy of the current steps, refining any errors, thinking of how to call SQL tools, and revisiting previous steps.
**Output Format (STRICTLY ENFORCED):**
- Conduct thinking inside `<think>...</think>` blocks every time you get new observation or information. Start with `<think>...</think>` blocks in your responses as shown in the following example.
- You can use SQL tool written within a single `<SQL>your SQL</SQL>` block to explore or verify. You can't use the format ```SQL ; \n```, you must use the format `<SQL>your SQL</SQL>` to get the output. `<SQL>your SQL</SQL>` block should follow closely behind `<think>...</think>` block. SQL tool output will be shown as dataframe inside `<observation>...</observation>`. Based on this observation, you can think again and refine.
- The returned dataframe will be truncated in 50 rows if observation is too long.
- If you find no further exploration is needed or have only 1 turn left, you MUST directly provide the final SQL query solution inside `<solution>...</solution>`.
- All your responses should be in the `<think>...</think>`, `<sql>...</sql>`, `<observation>...</observation>`, `<solution>...</solution>` blocks.

**Example:**
**Question:** how many pigs are in the farm?
**Database Schema:**
Table: animals
- id (INTEGER, PRIMARY KEY)
- species (TEXT)
- age (INTEGER)
- name (TEXT)
**Output:**
```
<think>I am querying how many pigs are in the farm.  I will begin
by checking if the 'animals' table exists and contains entries
with species = 'pig'.</think>
<SQL>SELECT COUNT(*) FROM animals WHERE species = 'pig';</SQL>
<observation>
+----------+
| COUNT(*) |
+----------+
| 12 |
+----------+
</observation>
<think>The result indicates that there are 12 pigs in the farm.
Since the question asks for how many pigs, I can now output the
final SQL as the solution.</think>
<solution>SELECT COUNT(*) FROM animals WHERE species =
'pig';</solution>
```

26

Table 18: Prompt example for SQL reasoning without memory.

Table 19: Ablation study on the maximum number of interaction turns ($T$). We compare the original model against models fine-tuned with different turn limits. For each setting, we report **Greedy** accuracy (from a single output), **Best of 8** accuracy, and **Selected** accuracy (after reranking 8 candidates) with grounding agent. All scores are execution accuracy (%) on the BIRD-dev benchmark.

| Model (Trained w/ Max Turns) | Inference (T=1) | | | Inference (T=5) | | | Inference (T=10) | | |
|---|---|---|---|---|---|---|---|---|---|
| | Greedy(%) | Best of 8(%) | Select(%) | Greedy(%) | Best of 8(%) | Select(%) | Greedy(%) | Best of 8(%) | Select(%) |
| Original Model (Base) | 54.43 | 77.76 | 69.69 | 55.41 | 77.82 | 70.34 | 55.76 | 77.56 | 70.07 |
| Trained (T=1) | 66.41 | 78.6 | 72.06 | 66.95 | 78.76 | 72.75 | 67.60 | 80.63 | 74.19 |
| Trained (T=5) | 67.60 | 82.19 | 75.29 | 69.30 | 83.7 | 77.84 | 68.25 | 82 | 76.40 |
| Trained (T=10) | 67.73 | 83.61 | 76.86 | 69.36 | 83.95 | 77.12 | 69.88 | 83.89 | 77.57 |

Table 20: Impact of "Best-of-N" selection on the BIRD-dev benchmark. **Greedy (Best of 1)** is the execution accuracy of the final selected trajectory. **Best of N** represents the upper-bound performance (Pass@N), indicating the percentage of times at least one correct trajectory was found among N candidates. Inference parameters: temperature=0.8, top_k=50, top_p=0.7, max_iterations=5.

| Selection Strategy | Execution Accuracy (%) |
|---|---|
| Greedy (Best of 1) | 69.30 |
| Best of 2 | 74.04 |
| Best of 4 | 79.71 |
| Best of 8 | 83.76 |
| Best of 16 | 86.31 |
| Best of 32 | 87.54 |

---

**Prompt for Generative Validation Agent**

**User:**

**Task Background:**
You are an expert SQL data analyst. Your task is to verify if a proposed solution correctly answers a user's question.
**Problem:**
{*question*}

**External Knowledge:**
{*external_knowledge*}

**Proposed Solution:**
{*solution_text*}

---

**Your Task:**
Based on all the information, is the SQL query in the solution logically correct for answering the question?
You must answer with "Yes" or "No" first, before any other text.

Is the answer correct (Yes/No)?

---

Table 21: The prompt used for the Generative Verifier. The agent is framed as a SQL expert and is provided with the problem, the proposed SQL query, and any relevant external knowledge. It evaluates the logical correctness of the query and must provide a final "Yes" or "No" judgment.

---

**Prompt for Selection Agent (LLM as a Judge)**

**User:**

**Task Background:**
You are an expert SQL data analyst. Your task is to select the BEST SQL query that correctly answers a user's question.
You are given several candidates. For each candidate, you will see its reasoning, the SQL query itself, and importantly, **the result of executing that query on the database.** A query might look correct but return an error or empty/wrong data. You must use the execution observation to make your final decision.
Here is the user's question:
{*question*}

Evaluate the following candidates based on ALL available information. Does the "Execution Observation" for a candidate actually answer the user's question?
—

{*formatted_candidates*}
—

**Final Analysis:**
Considering the reasoning, the SQL code, and especially the **execution results**, which single candidate provides the most correct and complete answer to the user's question?
**Instructions for your response:**

- Respond with ONLY the index number of the single best candidate.
- If multiple candidates produce correct results, select the one with the LOWEST index number.
- Do not include any other words, symbols, or explanations.

Best candidate index:

---

Table 22: The prompt used for the Selection Agent, operating as an LLM judge. It guides the model to select the best SQL query from multiple candidates by evaluating their reasoning, SQL code, and critically, their execution observations. Strict output instructions ensure a direct index selection.

Table 23: Ablation study of different selection strategies. The first row, **Pass@8**, shows the baseline execution accuracy (%) of the eight candidate trajectories from our Generator agent before any selection. Subsequent rows report the final accuracy after applying each method to select the best trajectory. **Self-Consistency** picks the most frequent result, **LLM as a Judge** uses GPT-4o/Qwen for selection, and **Ours** uses our fine-tuned 7B Generative Verifier.

| Method | Model Size | Bird dev (%) | Spider test (%) | Spider DK (%) |
|---|---|---|---|---|
| *Pass@8 (Generator Output)* | - | *83.76* | *90.68* | *82.06* |
| LLM as a Judge (GPT-4.1) | Unkonwn | 75.15 | 83.47 | 71.40 |
| LLM as a Judge (Qwen) | 7B | 70.47 | 79.60 | 70.09 |
| Self-Consistency | - | 72.93 | 83.51 | 73.08 |
| **Ours (Generative Verifier)** | **7B** | **77.84** | **89.75** | **78.13** |

