# OpenReview forum: "MARS-SQL: A Multi-Agent Reinforcement Learning Framework for Text-to-SQL"
_ICLR.cc/2026/Conference — Submitted to ICLR 2026_

### Official Review · Reviewer_m4x3 · 2025-10-25

**Soundness:** 2
**Presentation:** 3
**Contribution:** 1
**Rating:** 4
**Confidence:** 4

**Summary:**

This paper presents MARS-SQL, a multi-agent framework for Text-to-SQL. The system decomposes the task into three distinct stages: (1) a Grounding Agent for reasoning-driven schema linking, (2) a Generation Agent trained with RL (specifically GRPO) to interactively build queries in a multi-turn, ReAct-style loop, and (3) a Validation Agent trained via SFT to select the best query from multiple generated trajectories. The authors report SOTA execution accuracy for a 7B parameter model on the BIRD-dev set. The model also demonstrates strong out-of-domain generalization on the Spider test set, despite being trained exclusively on the BIRD dataset.

**Strengths:**

- Most part of this paper is clearly written and easy to follow
- The proposed multi-agent methodology itself appear to be technically sound and presents a principled approach to managing the complexity of the NL2SQL task.
- The execution accuracy, if can be further verified by the BIRD test set, seems to be a strong result on the BIRD bench for 7B models.

**Weaknesses:**

- The contribution and novelty of this paper is limited. LLM-based schema linker + multi-turn generator + LLM verifier is a well-established NL2SQL pipeline, the subset of which, if not full, has been discussed in many prior papers. Vanilla GRPO and SFT is used for tuning the agents, which are also well known methods by the research community and industry.
- Missing important technical details and insights for the research community to learn and reproduce the results, especially on how to overcome the challenges of multi-turn RL fine-turning, why the synergy between schema linker and verifier have such huge accuracy gain, and the insights into generalizability from BIRD to Spider.
- Given the surprisingly strong results in BIRD-Bench (potentially top 1 on BIRD-bench leaderboard), it is essential to cross-validate the findings on the official BIRD test set, which is missing in this paper.

**Questions:**

- Given the surprisingly good results in this paper that surpass many much larger models, it will be beneficial to further include the official BIRD test results to cross-validate the generalizability of the methodology.
- Why does the grounding agent only perform table-level schema linking instead of the column-level?
- Is there any scenario in multi-turn generation where there is no error or empty results returned from database execution but the generator still decides not to finalize the answer?
- It is notoriously challenging to train LLMs with multi-turn RL. Can authors share any insights or lessons learnt from overcoming the fundamental challenges of multi-turn RL training, especially in NL2SQL tasks?
- What is the accuracy of the verifier on BIRD dev and Spider dev?
- Can authors provide more qualitative or quantitative analysis to explain *why* training on BIRD leads to SOTA performance on Spider. What specific, robust reasoning patterns did the agent learn from BIRD that transfer so effectively? Without this analysis, the extraordinary result is just a number on a table, offering no insight or lesson for the research community.
- From table 3, it looks like the grounder and verifier work extremely well when used together on BIRD-dev. However, the verifier contributes to the majority of the EX improvement on spider datasets. Can authors explain why? It looks counter-intuitive because the verifier was SFTed only on the BIRD train set and it doesn’t have schema information from the databases. How can it understand the nuances in schema and value linking errors?
- Can authors clarify the setting for “Generator Only (Base)” in table 3? Is it using the base model or simply RL tuned generator as a baseline?
- Can the pipeline generalize to more challenging datasets, such as spider 2?

---

> ### Author Response · Authors · 2025-11-20
> **(1/7) Responses by Authors**
>
> **We have marked the added content in blue in the revised paper. Thank you again for your careful reading and valuable review.**
> > W1 The contribution and novelty of this paper is limited. LLM-based schema linker + multi-turn generator + LLM verifier is a well-established NL2SQL pipeline, the subset of which, if not full, has been discussed in many prior papers. Vanilla GRPO and SFT is used for tuning the agents, which are also well known methods by the research community and industry.
>
> Thanks for the feedback, we respectfully but strongly disagree with the comment reduces our work to a “well-established linker + generator + verifier pipeline with vanilla GRPO/SFT”. This overlooks the actual technical contributions. The core novelty lies in **(1) *how* our multi-agent system is trained** and **(2) the *novel application* of RL to a stateful, multi-turn environment.**
>
> ### 1. Innovation: A Trained Multi-Agent System vs. a Prompt-based Workflow
>
> Table 1 already summarizes how MARS-SQL differs from prior “multi-agent” designs; here we emphasize that our focus is on a **trained multi-agent system**, not a prompt script. The reviewer points to the high-level “Linker + Generator + Verifier” pipeline, which is indeed a natural abstraction. However, many prior works (e.g., XiYan-SQL) instantiate this as a **non-trained, prompt-based workflow** that simply chains calls to a single powerful closed-source LLM (e.g., GPT-4).
>
> In several other domains, there is a similar shift from prompt-orchestrated workflows to **trained agents**—for example in code agents (SWE-Dev[1] / SWE-Gym[2]), web agents (WEBRL[3] / WebAgent-R1[4] / ScribeAgent[5]), and deep-research agents (Tongyi DeepResearch[6] / WebDancer[7]). Our work pushes this trend further in the Text-to-SQL setting: we introduce a **multi-agent training framework** where the linker, generator, and verifier are all **learned policies within a shared RL formulation**, rather than a single fine-tuned agent plugged into a fixed, hand-crafted workflow.
>
> Our contribution is conceptually different:
>
> * We build a **fully trained multi-agent system** composed of smaller, specialized open models (7B), rather than a prompt chain over a monolithic API model.
> * The “pipeline” is a **learned communication protocol** between distinct agents: the Grounder *learns* (via RL) to output an optimal pruned schema for the Generator; the Generator *learns* (via RL) to drive an interactive trajectory for the Verifier and DB.
> * Our central claim is that this **trained, specialized multi-agent architecture** is strictly more effective and data-efficient than a black-box prompt workflow.
>
> ### 2. Innovation: Stateful Multi-Turn Interactive RL vs. "Vanilla" Stateless RL
>
> Our framework does **not** use “vanilla” GRPO. The key difference is that we move from a **single-turn, bandit-style** formulation to a **stateful, multi-turn RL** formulation.
>
> * Prior work (e.g., Arctic-Text2SQL-R1) applies RL (like GRPO) in a **stateless, single-turn** setting. The agent's task is analogous to a contextual bandit: it performs *one action* (generates the *entire* SQL query $y$) given the context ($x$). The policy is $\pi(y|x)$, and it receives one reward at the end.
> * Our work introduces a **stateful, multi-turn, interactive** RL framework. Our Generation Agent is a true RL agent operating in a sequential decision-making process.
>
>   * Its policy is **stateful**: $\pi(a_t | s_t)$, where the state $s_t$ includes the history of all previous `Think-Act-Observe` steps.
>   * Its actions $a_t$ are **interactive**: It executes a query against a live database.
>   * Its next state $s_{t+1}$ is determined by the **live database feedback** ($o_t$).
>
> This shift from single-turn to **multi-turn / interactive RL** is consistent with recent trends in LLM agent RL. This shift to **multi-turn, interactive RL** formulations have also been widely discussed in recent works like**MAGRPO** [8] and **AReaL** [9].
>
> [1] SWE-Dev: Evaluating and Training Autonomous Feature-Driven Software Development
>
> [2] Training Software Engineering Agents and Verifiers with SWE-Gym
>
> [3] WebRL: Training LLM Web Agents via Self-Evolving Online Curriculum Reinforcement Learning
>
> [4] WebAgent-R1: Training Web Agents via End-to-End Multi-Turn Reinforcement Learning
>
> [5] ScribeAgent: Towards Specialized Web Agents Using Production-Scale Workflow Data
>
> [6] Tongyi DeepResearch Technical Report
>
> [7] WebDancer: Towards Autonomous Information Seeking Agency
>
> [8] LLM Collaboration With Multi-Agent Reinforcement Learning
>
> [9] AReaL: A Large-Scale Asynchronous Reinforcement Learning System for Language Reasoning

---

> ### Author Response · Authors · 2025-11-20
> **(2/7) Responses by Authors**
>
> > W2 Missing important technical details and insights for the research community to learn and reproduce the results, especially on how to overcome the challenges of multi-turn RL fine-turning, why the synergy between schema linker and verifier have such huge accuracy gain, and the insights into generalizability from BIRD to Spider.
>
> Thanks for this valuable feedback. We provide the corresponding technical details and insights below:
>
> ### 1. How Multi-Turn Execution Works
>
> In short, each RL step is one live DB interaction plus its observation, and the policy learns when to issue another <sql> call and when to stop with <answer>.
>
> | Element | Description |
> | :--- | :--- |
> | **State ($s_t$)** | The current context, including the original question and the entire history of previous ``, ``, and `` tags. |
> | **Action ($a_t$)** | The agent outputs the next action, contained within the `...` tag. |
> | **Execution** | We use a **regular expression** to reliably extract the SQL code from the agent's output $a_t$. This extracted code is then **executed directly against the live database**. |
> | **Observation ($o_t$)** | The environment returns the raw database execution result or an error message. |
> | **New State ($s_{t+1}$)** | The agent processes $o_t$ and transitions to a new state. The model *always* performs an internal thought process. **If it needs a new round, it generates another `...` tag** ($a_{t+1}$). **If reasoning is complete, it outputs the terminal tag `...`**, which stops the rollout without further execution. |
>
> ### 2. Grounding Agent Insight: Quantifying Complexity Reduction
>
> The Grounding Agent's value is not just in filtering, but in **quantifiably reducing the search space** (and thus the computational complexity) for the Generation Agent.
>
> Let $S_{db}$ be the full database schema (all available tables and columns).
> Let $S_{vocab}$ be the LLM's full output vocabulary.
> Let $S_{link}$ be the sparse, pruned schema output by the Grounding Agent.
>
> * **Complexity without Grounder (Baseline):** The Generator $G_{gen}$ must select tokens from $S_{vocab} \cup S_{db}$. The model is constantly penalized by the sheer number of irrelevant entities in $S_{db}$, slowing down inference and increasing the chance of hallucination. The complexity of the SQL generation task is effectively tied to $O(|S_{db}|)$.
> * **Complexity with Grounder (MARS-SQL):** The Grounder $G_{link}$ outputs the essential, high-quality, pruned schema $S_{link} \subset S_{db}$, where $|S_{link}| \ll |S_{db}|$. The Generation Agent only conditions its output on $S_{link}$.
> * **Insight:** The Grounder transforms the task from a **high-complexity, general search problem ($O(|S_{db}|)$)** into a **low-complexity, specialized generation problem ($O(|S_{link}|)$)**. This is why our final execution accuracy is so high—the Generator is focused on logic, not filtering noise.
>
> ### 3. Validation Agent (Verifier) Insight: Separating Generation from Selection
> The verifier's gain(genrm[10], openai process rm[11]) is not due to "symbolic analysis" but due to generative modeling: it is fine-tuned to model the probability of a correct output, enabling it to accurately pick the best candidate from the Generator's diverse pool. As shown in Table 11, line 1095-1105, the Generator is capable of solving 87.54% of queries, and the Verifier's job is to efficiently close that gap, which it does extremely well.
>
> **Impact of "Best-of-N" Selection on BIRD-dev**
> | Selection Strategy | Execution Accuracy (%) |
> |--------------------|------------------------|
> | Greedy (Best of 1) | 69.3                   |
> | Best of 2          | 74.04                  |
> | Best of 4          | 79.71                  |
> | Best of 8          | 83.76                  |
> | Best of 16         | 86.31                  |
> | Best of 32         | 87.54                  |
>
> [10]: Generative Verifiers: Reward Modeling as Next-Token Prediction
>
> [11]: Let’s Verify Step by Step

---

> ### Author Response · Authors · 2025-11-20
> **(3/7) Responses by Authors**
>
> > W3&Q1 Given the surprisingly strong results in BIRD-Bench (potentially top 1 on BIRD-bench leaderboard), it is essential to cross-validate the findings on the official BIRD test set, which is missing in this paper.
>
> We are currently going through the official BIRD test submission pipeline, and in the meantime we provide an additional generalization result on Sparc (85.78% execution accuracy) for the reviewer’s reference. We have not yet appeared on the BIRD leaderboard because our multi-agent pipeline (especially the Generation Agent with live DB interaction) requires careful engineering to integrate with the leaderboard environment.
> In addition, we evaluate MARS-SQL on the out-of-domain Sparc benchmark, where it achieves 85.78% execution accuracy, substantially surpassing strong recent baselines and demonstrating robust generalization to sequential, context-dependent dialogues.
>  **OOD Generalization Results (Zero-Shot)**
> | Model  | Spider-test|Spider-DK | Sparc |
> | :--- | :---: | :---: | :---: |
> | GPT-O3-mini | 78.82%| 71.77%| **67.0%** |
> | GPT-5 | 77.1%|66.92% | **61.8%** |
> | Qwen2.5-7B |75.87% | 61.31%| **64.1%** |
> | **MARS-SQL (Ours)** | **89.75%**| **78.13%**| **85.78%** |
>
> >  Q2 Why does the grounding agent only perform table-level schema linking instead of the column-level?
>
> A column-level design(one LLM call per column) would dramatically increasing latency and token cost. This table-level option is a deliberate design choice for accuracy–efficiency trade-off. Our Grounding Agent works at the table level because each call already sees the full schema of a table (all columns, types, descriptions, examples). In a single pass it can (1) decide whether the table is relevant and (2) pick the needed columns inside that table. In practice, this is sufficient for accurate grounding (97.78% recall in Table 8). Table-level grounding therefore gives us almost the same utility as column-level at a fraction of the computational cost.
>
> > Q3 Is there any scenario in multi-turn generation where there is no error or empty results returned from database execution but the generator still decides not to finalize the answer?
> Yes, absolutely. This is a core feature of our agent's learned reasoning policy. The agent does not just check for *errors*; it evaluates the *semantic content* and *plausibility* of the observation against its multi-step plan.
>
> There are two common scenarios where a non-error, non-empty result does not end the process:
>
> ### 1. Complex Problem Decomposition
>
> When a problem is complex, the agent first forms a multi-step plan. A successful query execution may only be "Step 1" of that plan (e.g., finding an intermediate value). The agent receives the valid result, incorporates it into its reasoning, and continues to the next step.
>
> **Example: Decomposing a query to find an average and then filter by it.**
>
> > **Question:**
> > Consider the average difference between K-12 enrollment and 15-17 enrollment of schools that are locally funded, list the names and DOC type of schools which has a difference above this average.
> >
> > **Turn 1 (Think):**
> > To solve this problem, I need to follow these steps:
> >
> > 1. Calculate the average difference between K-12 enrollment and 15-17 enrollment for all schools.
> > 2. Filter the schools based on the funding type.
> > 3. Calculate the difference between K-12 enrollment and 15-17 enrollment for the filtered schools.
> > 4. Identify the schools with a difference above the average.
> > 5. List the names and DOC types of these schools.
> >
> > **Turn 1 (SQL):**
> >
> > ```sql
> > SELECT AVG(Enrollment (K-12) - Enrollment (Ages 5-17)) AS avg_diff FROM frpm;
> > ```
> >
> > **Turn 1 (Observation):**
> > `16.261833`
> >
> > **Turn 2 (Think):**
> > **(The agent does not stop, as this is only Step 1.)**
> > The average difference... is approximately 16.26. Now, I will filter the schools based on the funding type (locally funded = 1) and calculate the difference for these schools.
> >
> > **Turn 2 (SQL):**
> >
> > ```sql
> > SELECT T2.School, T2.DOC FROM frpm AS T1 JOIN schools AS T2 ON T1.CDSCode = T2.CDSCode
> > WHERE T2.FundingType = 'Locally funded' AND (T1.`Enrollment (K-12)` - T1.`Enrollment (Ages 5-17)`) > 14.92;
> > ```
> >
> > **Turn 2 (Observation):**
> > `[('Circle of Independent Learning', '54'), ('Learning Community Charter', '00')]`
> >
> > **Turn 3 (Think):**
> > The query executed successfully and returned non-empty results. I will combine the two steps and write the final answer.
> >
> > **Turn 3 (Solution):**
> >
> > ```sql
> > SELECT T2.School, T2.DOC FROM frpm AS T1 INNER JOIN schools AS T2 ON T1.CDSCode = T2.CDSCode WHERE T2.FundingType = 'Locally funded' AND (T1.`Enrollment (K-12)` - T1.`Enrollment (Ages 5-17)`) > (SELECT AVG(T3.`Enrollment (K-12)` - T3.`Enrollment (Ages 5-17)`) FROM frpm AS T3 INNER JOIN schools AS T4 ON T3.CDSCode = T4.CDSCode WHERE T4.FundingType = 'Locally funded')
> > ```

---

> ### Author Response · Authors · 2025-11-20
> **(4/7) Responses by Authors**
>
> (Continued)
>
> > Q3 Is there any scenario in multi-turn generation where there is no error or empty results returned from database execution but the generator still decides not to finalize the answer?
>
> ### 2. Data Plausibility and Sanity Checking
>
> The agent's policy learns to be "suspicious." If it receives a non-empty, non-error result that seems *implausible* (e.g., enrollment of "1.0"), it will trigger a self-correction or investigation loop to verify the data before finalizing a potentially incorrect answer.
>
> **Example: Agent sanity-checks an "unreasonable" enrollment number.**
>
> > **Question:**
> > Which cities have the top 5 lowest enrollment number for students in grades 1 through 12?
> >
> > **Turn 1 (Think):**
> > I need to find the cities with the top 5 lowest enrollment numbers...
> >
> > **Turn 1 (SQL):**
> >
> > ```sql
> > SELECT T2.City, T1.`Enrollment (K-12)` FROM frpm AS T1 INNER JOIN schools AS T2 ON T1.CDSCode = T2.CDSCode ORDER BY T1.`Enrollment (K-12)` ASC LIMIT 5;
> > ```
> >
> > **Turn 1 (Observation):**
> > `[('Markleeville', 1.0), ('Oakland', 1.0), ('Palermo', 1.0)]`
> >
> > **Turn 2 (Think):**
> > **(The agent does not stop. It "thinks" the result is suspicious.)**
> > The result shows... enrollment numbers are very low (1.0), which might indicate an error in the data or the need to filter out non-enrollment entries. Let's check the data to ensure accuracy.
> >
> > **Turn 2 (SQL):**
> >
> > ```sql
> > SELECT CDSCode FROM frpm WHERE `Enrollment (K-12)` != 1.0 LIMIT 10;
> > ```
> >
> > **Turn 2 (Observation):**
> > `[('01100170130419'), ('01100170112607'), ('01100170109835'), ...]`
> >
> > **Turn 3 (Think):**
> > The `CDSCode` values show that there are multiple records with an enrollment of 1.0. This could indicate non-enrollment entries. We need to filter out these records to get accurate city names.
> >
> > **Turn 3 (SQL):**
> >
> > ```sql
> > SELECT T2.City FROM frpm AS T1 INNER JOIN schools AS T2 ON T1.CDSCode = T2.CDSCode WHERE T1.`Enrollment (K-12)` != 1.0 ORDER BY T1.`Enrollment (K-12)` ASC LIMIT 5;
> > ```
> >
> > **Turn 3 (Observation):**
> > `[('Bishop'), ('Kingsburg'), ('Oakland'), ('Williams'), ('Compton')]`
> >
> > **Turn 4 (Think):**
> > Now, the cities with the lowest enrollment numbers... are identified... We can now provide the final SQL query solution.
> >
> > **Turn 4 (Solution):**
> >
> > ```sql
> > SELECT T2.City FROM frpm AS T1 INNER JOIN schools AS T2 ON T1.CDSCode = T2.CDSCode WHERE T1.`Enrollment (K-12)` != 1.0 ORDER BY T1.`Enrollment (K-12)` ASC LIMIT 5;
> > ```
>
> In both cases, the agent's internal policy—not just the database—determines when the reasoning process is truly complete. We will add these 2 examples in our revised version.
>
> > Q4. Can authors share any insights or lessons learnt from overcoming the fundamental challenges of multi-turn RL training, especially in NL2SQL tasks?
>
> This is an excellent point, and the reviewer is correct—it was the most challenging part of this work. We are happy to share the key lessons we learned.
>
> The two fundamental challenges were **(1) generating high-quality, diverse trajectories** for the agent to learn from, and **(2) stabilizing the RL update step** to ensure the agent actually learns from that data. We overcame them with two key insights:
>
> ### 1. Lesson 1: Trajectory Generation and Max-Turn Tuning ($T$)
>
> The RL agent learns from a "buffer" of trajectories. If this buffer is "boring" (e.g., all trajectories are simple and correct, or all are complex and fail), the agent learns nothing. We found the **maximum turn limit ($T$)** was the most critical hyperparameter for controlling this.
>
> * We experimented with $T=1$, $T=5$, and $T=10$.
> * **$T=1$ (Single-Turn):** This setting failed. It is just a single-turn RL (like Arctic-Text2SQL) and is insufficient for teaching the agent multi-step reasoning.
> * **$T=10$ (Too High):** This was also suboptimal. It created two problems: (1) **Inefficiency:** It produced extremely long trajectories, slowing down training and inference. (2) **Context Limit:** The agent's reasoning would often "get lost" or hit context length limits on very complex queries, leading to low-quality, failed trajectories.
> * **$T=5$ (The "Sweet Spot"):** We found $T=5$ provided the best balance. It was long enough to allow the agent to generate a **rich and diverse set of trajectories**—some succeeding in 2 turns, some in 5, some failing after 3—but short enough to remain computationally feasible and maintain reasoning coherence.
>
> **Insight:** The goal is not to have the *longest* trajectories, but the most *diverse* set of reasoning paths. A well-tuned $T=5$ achieved this.

---

> ### Author Response · Authors · 2025-11-20
> **(5/7) Responses by Authors**
>
> (Continued)
> > Q4. Can authors share any insights or lessons learnt from overcoming the fundamental challenges of multi-turn RL training, especially in NL2SQL tasks?
>
> ### 2. Lesson 2: Modifying the GRPO Algorithm for Stability
>
> We used GRPO, but "vanilla" GRPO was not stable. The key challenge is that in a multi-turn setting, the reward difference (advantage) between a 2-turn correct answer and a 5-turn failed answer can be massive. We made two crucial algorithm-level modifications:
>
> * **Effective Group Sampling:** Rather than taking all generated trajectories, we increased our `group sampling size` (i.e., generated more trajectories per prompt). We then **filtered this group, "throwing away" (discarding) low-quality training batches** where all trajectories were too similar (e.g., all failed, or all succeeded with minor variations). This ensures the policy gradient is calculated on meaningful pairs with a clear "winner" and "loser."
> * **Relaxing the Clipping Constraint (The "Epsilon Trick"):** The standard GRPO/DAPO clip ($\epsilon$) is designed for stability, but we found it was too restrictive. It would "clip" the advantage from a brilliant, novel 2-turn solution, preventing the agent from learning *how good* that solution was.
>
>   * **Our Modification:** We implemented a dynamic clipping mechanism. When the absolute advantage ($|Adv|$) was **very high (i.e., $|Adv| > \epsilon$)**, we would **relax or remove the clipping constraint**.
>   * **Why it works:** This allowed the policy to take a much larger "step" when it stumbled upon a significantly better (or worse) reasoning path. This was crucial for escaping local optima and aggressively rewarding the agent for discovering novel, efficient, and correct solutions.
>
> **In summary, our key lessons were:**
>
> 1. Carefully curate the training data (trajectories) by tuning the max turn limit ($T=5$).
> 2. Aggressively filter the training batches ("group sampling").
> 3. Modify the RL update rule (relaxing the $\epsilon$-clip) to allow the agent to learn more from high-impact "Aha!" moments.
>
> We will add these insights to the Appendix.
>
> > Q5. What is the accuracy of the verifier on BIRD dev and Spider dev?
>
> Our accuracy result is in Table 14 (line lines 1174-1185)
>
> | Method | Model Size | Bird dev (%) | Spider test (%) | Spider DK (%) |
> | :--- | :--- | :---: | :---: | :---: |
> | *Pass@8 (Generator Output)* | - | *83.76* | *90.68* | *82.06* |
> | LLM as a Judge (GPT-4.1) | Unknown | 75.15 | 83.47 | 71.40 |
> | LLM as a Judge (Qwen) | 7B | 70.47 | 79.60 | 70.09 |
> | Self-Consistency | - | 72.93 | 83.51 | 73.08 |
> | **Ours (Generative Verifier)** | **7B** | **77.84** | **89.75** | **78.13** |
>
> And the accuracy of the verifier can be calculated based on the result:
> | Benchmark | Our Final Accuracy | Pass@8 (Oracle) | Verifier Selection Accuracy |
> | :--- | :---: | :---: | :---: |
> | **BIRD dev** | 77.84% | 83.76% | **92.93%** |
> | **Spider test** | 89.75% | 90.68% | **99.08%** |
> | **Spider DK** | 78.13% | 82.06% | **95.21%** |

---

> ### Author Response · Authors · 2025-11-20
> **(6/7) Responses by Authors**
>
> > Q6. Can authors provide more qualitative or quantitative analysis to explain why training on BIRD leads to SOTA performance on Spider.
>
> This is an excellent and insightful question. The reviewer is correct that this result (SOTA on Spider via zero-shot transfer from BIRD) is a key claim and demands a deeper explanation.
>
> Our core argument is that BIRD, being a *significantly harder and "noisier"* dataset, acts as a "challenge" training ground. It forces our agent to learn **general-purpose reasoning policies** via its multi-turn RL interaction, which are then *over-qualified* for the simpler, cleaner Spider environment.
>
> The two most critical, transferable patterns our agent learns are:
>
> 1. **Interactive Problem Decomposition:** Many BIRD queries are too complex for a single-pass `SELECT`. Our agent learns a general policy to break them down:
>
>     * It **thinks** of a multi-step plan.
>     * It **acts** by executing a query to find an intermediate value (e.g., an `AVG` or `COUNT`).
>     * It **observes** the numerical result.
>     * It **thinks** again, using this new value to construct the *final* query.
>       This `Think-Act-Observe` loop for decomposition is a *general reasoning skill* that is not specific to BIRD's schema. When applied to Spider's most complex queries (e.g., those with nested subqueries), our agent can solve them robustly, whereas SFT models fail.
> 2. **Robustness to Schema Imperfection:** BIRD's schemas are notoriously "dirty" and "noisy." Our agent learns (through RL and interaction) that its initial schema linking might be incomplete. It learns a policy of **"distrust and verify,"** using its `Act` turn to run exploratory queries (e.g., re-querying column info) if its first attempt fails.
>
>     * This makes the agent incredibly robust. When it encounters a clean, simple Spider database, this "robustness" policy simply has an easier job. It doesn't fail on minor ambiguities that might confuse a brittle SFT model.
>
> In short, training on BIRD's complex, messy data with our interactive RL framework forces the agent to learn *how to problem-solve in general*. When this highly-skilled agent is given the "easier" Spider test, it excels. This is why our model achieves SOTA on Spider without ever being trained on it, and it provides a clear lesson for the community: **training on complex, interactive tasks (like BIRD) is a highly effective way to forge general, robust reasoning skills.**
>
> >Q7. From table 3, it looks like the grounder and verifier work extremely well when used together on BIRD-dev.
>
> To clarify why the verifier contributes more on **Spider** despite being SFTed only on **BIRD**, we highlight a key difference between the two datasets: **Spider queries and databases are significantly simpler**. This means that even without explicit schema tokens or DB-specific knowledge, the verifier can still make reliable judgments based on its inputs (question, trajectory, predicted SQL, and execution result).
> Below is an illustrative comparison of two datasets:
> | Dataset | Avg. Schema Token Length | Avg. #Columns per DB | Avg. #Toknes/SQL |
> |---------|---------------------------|------------------------|----------------|
> | **BIRD-dev** | ~950 | 54.56 | 18.5|
> | **Spider** | ~250 | 26.82 | 30.9 |
> Spider’s much smaller schema space reduces the burden on schema linking. So grounding agent contributes little to spider tasks, and the verifier contributes to the majority of the EX improvement on spider.
> ### Why the Verifier Still Works Without Schema Information
> The verifier’s input during inference includes:
> - The **full trajectory** produced by the Generation Agent
> - The **natural language question**
> - The **predicted SQL query**
> - The **execution result of the predicted SQL query**
> Because these inputs already encode the necessary semantic and structural cues, the verifier does **not** need explicit schema tokens. It simply needs to judge whether the predicted SQL is *consistent* with the question and whether the execution result indicates a potential mismatch. Thus, the reviewer’s concern about lacking schema knowledge does not apply in our setting.
>
> > Q8. Clarification for “Generator Only (Base)” in table 3.
>
> Thank the reviewer for the clarification question. "Generator Only (Base)" is an ablation setting that uses **only our RL-tuned Generation Agent** in isolation.
>
> In this specific baseline:
> * The **Grounding Agent is disabled** (the generator receives the full, un-pruned database schema).
> * The **Validation Agent is disabled** (the generator outputs only one trajectory, $N_g=1$, which is taken as the final answer).
>
> This setting is designed to measure the standalone performance of the core multi-turn Generation Agent, without the benefits of the other two agents.

---

> ### Author Response · Authors · 2025-11-20
> **(7/7) Responses by Authors**
>
> > Q9. Can the pipeline generalize to more challenging datasets, such as spider 2?
>
> Our design is **fully compatible** with Spider 2.0: the same multi-agent, multi-turn RL pipeline can in principle operate on larger schemas and more complex queries, and we are actively exploring this direction. At the current stage, the bottlenecks are **practical**, not conceptual: (1) collecting enough training data for the Grounding Agent on **much larger, more complex databases**, and (2) the **base model’s ability**[12,13,14] to handle longer contexts and multi-turn trajectories required by Spider 2.0. Addressing these data and model-capacity issues is an exciting direction for future work, but is orthogonal to the core multi-turn reasoning contributions of this paper.
>
> [12]: ReFoRCE: A Text-to-SQL Agent with Self-Refinement, Consensus Enforcement, and Column Exploration
>
> [13]: AgenticData: An Agentic Data Analytics System for Heterogeneous Data
>
> [14]: RSL-SQL: Robust Schema Linking in Text-to-SQL Generation

---

> ### Author Response · Authors · 2025-11-26
>
> Dear Reviewer m4x3,
>
> We deeply appreciate the time and effort you dedicated to providing such a comprehensive and detailed review. We would like to follow up to ensure we have effectively addressed your concerns regarding novelty, technical mechanics, and generalization.
>
> Given your extensive list of questions, we have structured our response into three key areas:
> 1. Clarified Novelty & Contribution (W1): We explicitly distinguished MARS-SQL from "vanilla" pipelines. We highlighted that our core contribution is a trained (vs. prompt-based) and stateful, multi-turn RL (vs. single-turn) framework, which represents a fundamental shift in how Text-to-SQL agents are optimized.
>
> 2. Provided Deep Technical Insights (W2, Q1-Q4, Q8): To aid reproducibility, we detailed the crucial role of trajectory diversity (optimal $T=5$) and our "dynamic clipping" modification to stabilize GRPO. For agent behaviors, we detailed the adaptive stopping rule (the agent learns when to stop, not just fixed turns) and the efficiency behind table-level grounding.
>
> 3. Validated Generalization & Robustness (W3, Q5-Q7, Q9): We added results on Sparc (85.78%), verifying OOD generalization beyond Spider. We analyzed the "BIRD-to-Spider" transfer, explaining how learning interactive problem decomposition on complex BIRD queries naturally transfers to simpler tasks. We also reported the Verifier accuracy (~93-99%) to explain its synergy with the Generation Agent.
>
> We hope these detailed clarifications and additional analyses fully resolve your concerns. We kindly ask if you would consider re-evaluating our work based on these new insights.
>
> Thank you very much for your time and attention.

---

### Official Review · Reviewer_15cV · 2025-10-31

**Soundness:** 2
**Presentation:** 3
**Contribution:** 2
**Rating:** 2
**Confidence:** 5

**Summary:**

This paper introduces MARS-SQL, a multi-agent reinforcement learning framework designed to improve Text-to-SQL task. The authors model Text-to-SQL as a multi-agent decision process involving three specialized agents: Grounding Agent identifies relevant tables and columns from the schema using GRPO; Generation Agent employs an interactive ReAct-style reasoning loop to generate SQL queries that adapt to database feedback; Validation Agent evaluates multiple candidate SQLs and selects the best one through a learned judgment process. Extensive experiments on BIRD, Spider, and Spider-DK benchmarks demonstrate new state-of-the-art execution accuracy among open-source systems. Ablation and scaling studies confirm the contribution of each agent and the benefits of multi-round interaction.

**Strengths:**

1. The Grounding Agent is trained with GRPO to optimize schema selection via explicit reward feedback, rather than relying solely on supervised matching heuristics. This RL-based grounding approach allows the model to dynamically balance precision and recall of relevant tables and columns, improving robustness on complex database schemas.
2. The Generation Agent integrates a multi-turn reasoning–action–feedback loop, where each SQL generation step is conditioned on execution feedback from the database.
3. Experiments on BIRD-dev, Spider-test show that MARS-SQL consistently outperforms prior state-of-the-art open-source systems,

**Weaknesses:**

1. The overall system design closely resembles XiYan-SQL, following a multi-agent pipeline with schema linking, SQL generation, and selection modules. Moreover, the use of GRPO for training the Text-to-SQL model has already been explored in Arctic-Text2SQL-R1. The conceptual innovation beyond these prior frameworks is limited, and the contribution mainly lies in engineering refinement rather than new theoretical insights.
2. The framework requires training two RL-based agents and one SFT verifier, making it resource-intensive. The paper does not quantify training time, or compute cost, nor does it compare against non-trained multi-agent systems (e.g., prompt-based approaches).
3. The paper benchmarks primarily against Arctic-Text2SQL-R1, which is a single-model system. Comparing a multi-agent system to a single-agent baseline underrepresents the true performance gap. A fairer comparison should include multi-agent baselines, such as entries on the BIRD leaderboard or Spider 2.0-Snow benchmark.
4. The abstract and discussion emphasize that RL training improves robustness, especially in schema linking. However, the paper provides no dedicated experiment or metric explicitly evaluating robustness (e.g., under schema perturbations, synonym replacements, or unseen database structures). As a result, the robustness claim lacks empirically supported.

**Questions:**

1. Could you clarify how many RL training steps were used for the Grounding Agent and Generation Agent respectively? Additionally, how did the reward and rollout length evolve during training? Were there stability or convergence patterns observed?
2. For the multi-turn SQL Generation Agent, how did the actual number of reasoning turns used during training vary across examples? Beyond the maximum turn limit T, was there any minimum turn constraint or adaptive stopping rule applied?

---

> ### Author Response · Authors · 2025-11-20
> **(1/4) Responses by Authors**
>
> **We have marked the added content in blue in the revised paper. Thank you again for your careful reading and valuable review.**
> > W1. The overall system design closely resembles XiYan-SQL, following a multi-agent pipeline with schema linking, SQL generation, and selection modules ...
>
> Our contribution goes beyond an incremental engineering refinement. MARS-SQL introduces a **trained multi-agent architecture** and a **stateful, multi-turn RL policy** that learns to interact with the database, reason over trajectories, and self-correct—moving beyond both prompt-only pipelines (XiYan-SQL) and single-turn RL-our contribution goes beyond an incremental engineering refinement. MARS-SQL introduces a trained multi-agent architecture and a stateful, multi-turn RL policy that learns to interact with the database, reason over trajectories, and self-correct—moving beyond both prompt-only pipelines (XiYan-SQL) and single-turn RL-tuned models (Arctic-Text2SQL).
>
> 1. **Trained Agents, Not a Prompt Workflow**: First, compared with XiYan-SQL, the key distinction is whether the “agents” are trained models or just prompts. XiYan-SQL is a non-trained, prompt-based workflow where different “agents” are simply different prompts to a single closed-source LLM (e.g., GPT-4). In contrast, MARS-SQL is a trained multi-agent system: our Grounder, Generator, and Validator are three specialized 7B open-source models, each fine-tuned for its own role. This shows that smaller, specialized, trained agents, jointly optimized with RL, can outperform a black-box prompt pipeline—shifting from prompt engineering to multi-agent system design.
>
> 2. **Multi-Turn (Stateful) RL, Not Single-Turn (Stateless) RL**: Second, compared with Arctic-Text2SQL, the core difference lies in the RL formulation. Arctic-Text2SQL applies GRPO in a single-turn (stateless) setting: the model outputs one full SQL query per question, and the RL episode is a single action with a terminal reward, i.e., a contextual bandit. In MARS-SQL, the Generation Agent learns a multi-turn, interactive (stateful) policy. It produces a trajectory of Think–Act–Observe steps, where the policy π(a_t | s_t) adapts to intermediate feedback from live database execution. Optimizing such a stateful, adaptive policy in an interactive environment is substantially harder than tuning a stateless generator, and, to our knowledge, MARS-SQL is the first to apply this multi-turn RL policy within a multi-agent Text-to-SQL framework.
>
> | Dimension                                      | XiYan-SQL | Arctic-Text2SQL-R1 | MARS-SQL (Ours) |
> | :-------------------------------------------- | :------: | :----------------: | :-------------: |
> | **Core Dimensions**                           |          |                    |                 |
> | **Trained agents beyond prompting**           | ✗        | ✓                  | ✓               |
> | Multi-agent *system* (more than one agent)    | ✓       | ✗                  | ✓               |
> | **Multi-turn, stateful RL**                   | ✗        | ✗                  | ✓               |
> |                                              |          |                    |                 |
> | **Other System Properties**                   |          |                    |                 |
> | Agent specialization (different roles/models) | ✗        | ✗                  | ✓               |
> | Uses RL for optimization                      | ✗        | ✓                  | ✓               |
> | Interactive `Think–Act–Observe` with DB       | ✗        | ✗                  | ✓               |
>
> > W2. The framework requires training two RL-based agents and one SFT verifier, making it resource-intensive. The paper does not quantify training time, or compute cost, nor does it compare against non-trained multi-agent systems (e.g., prompt-based approaches)
>
> Thanks for the suggestion. We have included the training cost below and some analysis to show why “resource-intensive” is reasonable.  For the comparison with non-trained multi-agent systems, the corresponding results are already reported in Table 2 (lines 329–345) in the main paper; we restate the key numbers here in the rebuttal for the reviewer’s convenience.
>
> 1) Training Time Cost Details: As shown, the entire specialized training for our multi-agent framework completes in **approximately 13 hours on 4 H800**. This is a very modest and practical one-time cost.
>
> **One-Time Training Cost on 4 NVIDIA H800**
>
> | Agent | Method | Training Steps | Batch Size | Est. Training Time |
> | :--- | :--- | :---: | :---: | :---: |
> | **Validation Agent** | SFT |  ~10k | 4 | 1 h |
> | **Grounding Agent** | GRPO RL | 600 steps | 64 | 4 h |
> | **Generation Agent** | GRPO RL | 160 steps | 64 | 8 h |
> | **Total** | | | | **13 hours** |

---

> ### Author Response · Authors · 2025-11-20
> **(2/4) Responses by Authors**
>
> > W2. The framework requires training two RL-based agents and one SFT verifier, making it resource-intensive. The paper does not quantify training time, or compute cost, nor does it compare against non-trained multi-agent systems (e.g., prompt-based approaches)
>
> `Response`: (Continued)
>
> 2) The proposed MARS-SQL is highly data-efficient: it reaches 77.84 EX with only 35k LLM-labeled examples, whereas single-agent SFT systems require orders of magnitude more supervised data and still underperform. This is because our multi-agent RL framework learns diverse reasoning and self-correction behaviors through interaction and self-play, rather than relying on ever-larger SFT datasets. We conservatively estimate that a "single-agent SFT-only system would require **≈15M LLM-synthetic examples** and **≈3–4 months** of training on the same hardware to match MARS-SQL’s **77.84 EX**.
>
> **Cost Analysis Compared with single-agent SFT-only system**
>
> | Method | Annotation | Data                 | Training time (wall)              | Dev EX |
> |--------|-----------|-----------------------------------------------|----------------------------------:|-------:|
> | **Original (baseline)** | —             | —                     | —                                | 54.56  |
> | **SFT on BIRD-training** | Human         | 12,000                  | **~2 h**                         | 61.08  |
> | **Large SFT (OminiSQL)** | LLM + Human   | 2,500,000                  | **~20 days**                     | 64.50  |
> | **MARS-SQL (ours)**      | LLM           | 35,000        | **~13 h**                        | **77.84** |
>
> 3) Our paper already made comparison against non-trained multi-agent systems in **Table 2(line lines 329-345)**. To make this crucial comparison clearer, we have extracted the most relevant ones below:
>
> **Restatement of key numbers compared with prompt-based methods**
>
> | Model | Source | Thinking / Agentic? | BIRD-dev (%) | Spider-test (%) |
> | :--- | :--- | :---: | :---: | :---: |
> | **Closed-Source Frameworks** | | | | |
> | CHESS | Closed | Yes | 65.00% | 87.20% |
> | XiYan-SQL | Closed | Yes | 73.34% | 89.65% |
> | CHASE-SQL + Gemini | Closed | Yes | 74.90% | 87.60% |
> | **MARS-SQL (Ours)** | **Open** | **Yes** | **77.84%}** | **89.75%** |
>
> > W3 The paper benchmarks primarily against Arctic-Text2SQL-R1, which is a single-model system. Comparing a multi-agent system to a single-agent baseline underrepresents the true performance gap. A fairer comparison should include multi-agent baselines, such as entries on the BIRD leaderboard or Spider 2.0-Snow benchmark.
>
> Our paper already made comparison against several multi-agent systems and reported in **Table 2(line lines 329-345)** (e.g. CHESS, XiYan-SQL, CHASE-SQL + GEMINI). We will add a column to show if the comparison group is multi-agent or not in the revised version. To make this comparison clearer, we restate the most relevant ones and add "Multi-Agent?" and "Trained?" column below:
>
> **Restatement of key numbers compared with multi-agent methods**
>
> | Model                | Multi-Agent? | Trained? | Thinking / Agentic? | BIRD-dev (%) | Spider-test (%) |
> | :------------------- | :----------: | :-------------------------: | :-----------------: | :----------: | :-------------: |
> | **Closed-Source Frameworks** |            |                           |                     |              |                 |
> | CHESS                | Yes          | No                          | Yes                 | 65.00%       | 87.20%          |
> | XiYan-SQL            | Yes          | No                          | Yes                 | 73.34%       | 89.65%          |
> | CHASE-SQL + Gemini   | Yes          | No                          | Yes                 | 74.90% | 87.60%    |
> | **Open-Source Frameworks**   |            |                           |                     |              |                 |
> | Arctic-Text2SQL-R1   | No           | Yes                         | Yes                 | 70.70%       | 88.80%          |
> | Reasoning SQL        | No           | Yes                         | Yes                 | 72.29%       | 81.43%          |
> | **MARS-SQL (Ours)**  | **Yes**      | **Yes**                     | **Yes**             | **77.84%** | **89.75%** |
>
> We will revise Table 2 in the revised version.

---

> ### Author Response · Authors · 2025-11-20
> **(3/4) Responses by Authors**
>
> > W4. The abstract and discussion emphasize that RL training improves robustness, especially in schema linking. However, the paper provides no dedicated experiment or metric explicitly evaluating robustness (e.g., under schema perturbations, synonym replacements, or unseen database structures). As a result, the robustness claim lacks empirically supported.
>
> **Already Included the Robustness Analysis on Schema**:  Our draft does report robustness evidence in Table 8. The grounding evaluation is explicitly designed to probe robustness under both schema perturbations and unseen databases. In particular, Spider-DK is a perturbed variant of Spider where schema element names are obfuscated or replaced with synonyms; the large gains from SFT to RL on Spider-DK (recall 88.60 → 98.13, precision 72.71 → 91.59) directly measure robustness of schema grounding under naming perturbations. Moreover, evaluating the same grounding model on both Spider and BIRD, which differ in schemas and domains, shows consistent improvements, demonstrating robustness to unseen databases.We restate the results from Table 8 here to help reviewer check the comparison. We will clarify this robustness interpretation of the grounding recall/precision metrics in the revised version.
>
> **(a) Robustness to schema perturbations (Spider → Spider-DK)**
>
> | Dataset   | Metric     | Base  | +SFT  | +RL (ours) | Δ (RL − SFT) |
> | :-------- | :--------- | :---: | :---: | :--------: | :----------: |
> | Spider    | Recall     | 87.48 | 90.39 | 98.97      | +8.58        |
> | Spider    | Precision  | 69.22 | 78.16 | 93.62      | +15.46       |
> | Spider-DK | Recall     | 84.25 | 88.60 | 98.13      | +9.53        |
> | Spider-DK | Precision  | 66.54 | 72.71 | 91.59      | +18.88       |
>
> **(b) Robustness to unseen databases (BIRD-dev ↔ Spider)**
>
> | Dataset  | Metric     | Base  | +SFT  | +RL (ours) | Δ (RL − SFT) |
> | :------- | :--------- | :---: | :---: | :--------: | :----------: |
> | BIRD-dev | Recall     | 68.59 | 74.97 | 97.78      | +22.81       |
> | BIRD-dev | Precision  | 53.45 | 67.01 | 90.74      | +23.73       |
> | Spider   | Recall     | 87.48 | 90.39 | 98.97      | +8.58        |
> | Spider   | Precision  | 69.22 | 78.16 | 93.62      | +15.46       |
>
>
> **Further Accuracy Verification on Sparc dataset**: Beyond schema-level grounding, we also assess end-to-end robustness on a different benchmark: on the out-of-domain, multi-turn Sparc dataset, our full MARS-SQL system achieves 85.78% execution accuracy, indicating that the learned reasoning and interaction policies transfer well to new databases and dialog-style interactions. We will clarify this robustness interpretation of both the grounding metrics and the Sparc results in the revised version.
>
> | Model | Exe. Acc. (%) |
> | :--- | :---: |
> | o3 | 57.0% |
> | GPT-4.1 | 61.8% |
> | Qwen3-14B | 64.1% |
> | **MARS-SQL (Ours)** | **85.78%** |
>
> > Q1. Could you clarify how many RL training steps were used for the Grounding Agent and Generation Agent respectively? Additionally, how did the reward and rollout length evolve during training? Were there stability or convergence patterns observed?
>
> **1. Training Steps and Convergence**
>
> Our RL training phases were efficient and targeted, building on already-strong SFT base models.
>
> * **Grounding Agent:** We trained for **600 steps** with a batch size of 64.
> * **Generation Agent:** We trained for **160 steps** with a batch size of 64.
>
> For both agents, we observed clear **stability and convergence patterns in the reward curves**. The reward consistently increased and then plateaued, indicating that the policies were successfully optimized within this relatively small number of steps.
>
> **2. Additional Analysis for Evolution of Rollout Length**
>
> In our setting, each `Think–Act–Observe` cycle corresponds to one database interaction turn, so the **average number of interaction turns** could serves as a trajectory-level proxy for rollout length.
>
> * **Grounding Agent(single-turn):**  the rollout length (token count) shows a mild **U-shaped** pattern: it first becomes more concise and then slightly lengthens to include only essential schema information, reflecting policy refinement rather than “emergent” reasoning from scratch.
>
> * **Generation Agent(multi-turn):** the average number of interaction turns consistently **decreases and stabilizes at a lower level**, meaning the agent both solves problems more directly and learns when to stop. This is quantitatively supported by the average DB interaction turns on BIRD-dev (max turns = 5):
>
> **Average Interaction Turns Before vs. After RL Training**
> *(On BIRD-dev, max turns = 5)*
> | Model | Avg. Turns (Challenging) | Avg. Turns (Moderate) | Avg. Turns (Simple) |
> | :--- | :---: | :---: | :---: |
> | **Before RL (Base Model)** | 2.90 | 2.67 | 2.27 |
> | **After RL (Generation Agent)** | **1.82** | **1.71** | **1.45** |
>
> We will add this discussion and figures about rollout length to the Appendix.

---

> ### Author Response · Authors · 2025-11-20
> **(4/4) Responses by Authors**
>
> > Q2: For the multi-turn SQL Generation Agent, how did the actual number of reasoning turns used during training vary across examples? Beyond the maximum turn limit T, was there any minimum turn constraint or adaptive stopping rule applied?
>
> Thanks for this excellent question.To answer the questions directly:
> 1.  **Minimum Turn Constraint:** No, there was **no minimum turn constraint** applied.
> 2.  **Adaptive Stopping Rule:** Yes, the agent employs a learned, **adaptive stopping rule**. It is not forced to continue until a maximum turn limit $T$.
>
> ### 1. Mechanism of Adaptive Stopping:
>
> The agent operates in a `Think-Act-Observe` loop. The "number of turns" in our statistics refers to the number of `Act-Observe` cycles (i.e., interactions with the database). The agent's reasoning process is as follows:
>
> 1.  The agent starts with an initial `<think>` step.
> 2.  It performs an `Act` (e.g., executing a query) and gets an `Observation`.
> 3.  It enters a new `<think>` step to analyze the feedback.
> 4.  **Crucially, within any `<think>` step, the agent's learned policy can decide that the task is complete and the SQL is correct.**
> 5.  When it reaches this conclusion, it emits the special **`<answer>`** tag, which immediately terminates the generation trajectory for that example.
>
> This design allows the agent to "self-discriminate." It can stop after just one interaction if the query is simple and the execution feedback is positive. Conversely, it can choose to continue the `Think-Act-Observe` loop for multiple turns if the problem is complex and requires iterative refinement.
>
> ### 2. Empirical Evidence of Adaptive Reasoning:
>
> - First, we see a strong correlation between query difficulty (as defined by the BIRD dataset) and the average number of turns the agent chooses to take. As shown, the agent correctly learns to apply more complex, multi-turn reasoning (1.82 turns) to "Challenging" queries, while efficiently solving "Simple" queries in fewer turns (1.45).
>
> **Average Interaction Turns by Query Difficulty (BIRD-dev)**
>
> | BIRD Difficulty | Avg. Interaction Turns |
> | :--- | :---: |
> | Simple | 1.45 |
> | Moderate | 1.71 |
> | Challenging | 1.82 |
>
>
> - Second, the overall distribution of turns shows that while the agent *can* use many turns, it solves the majority of problems efficiently. This distribution confirms that the agent is not wasteful. It solves the vast majority of problems (1116) in a single interaction. However, for the more difficult long-Ftail of queries, it has robustly learned to apply deeper reasoning (up to 5+ turns) to find the correct solution.
>
> **Distribution of Interaction Turns Used (BIRD-dev)**
>
> | # of Interaction Turns | # of Examples |
> | :--- | :---: |
> | 1 | 1116 |
> | 2 | 174 |
> | 3 | 105 |
> | 4 | 88 |
> | 5+ | 51 |
>
> In summary, our agent is not bound by arbitrary turn limits but learns an adaptive policy to decide how much reasoning is necessary for each specific query. We will add this analysis and these tables to the Appendix to clarify this important mechanism.

---

> ### Author Response · Authors · 2025-11-26
>
> Dear Reviewer 15cV:
>
> We deeply appreciate the time and effort you are dedicating to the review process. We would like to know whether we have addressed your concerns.
>
> Specifically, in our response we have:
>
> 1. Clarified Novelty (W1): detailed the fundamental difference of MARS-SQL as a trained (vs. prompt-based XiYan-SQL) and stateful/multi-turn RL (vs. stateless Arctic-Text2SQL) system.
>
> 2. Quantified Efficiency (W2): Reported the training cost (~13 hours on 4 H800s) and demonstrated superior data efficiency compared to SFT baselines.
>
> 3. Benchmarked Fairly (W3): Explicitly highlighted comparisons against existing multi-agent systems (e.g., XiYan-SQL, CHASE-SQL) in Table 2.
>
> 4. Verified Robustness (W4): Validated robustness against schema perturbations (Spider-DK) and OOD generalization (Sparc, 85.78%).
>
> 5. Detailed Dynamics (Q1/Q2): Provided training steps and empirical evidence of the agent's adaptive stopping mechanism.
>
> If you have any additional questions or require further clarification on any aspect of our work, please do not hesitate to let us know. We are more than happy to provide any additional information or address any concerns you may have.
>
> Thank you very much for your time and attention.

---

### Official Review · Reviewer_Km2H · 2025-11-01

**Soundness:** 3
**Presentation:** 3
**Contribution:** 3
**Rating:** 6
**Confidence:** 5

**Summary:**

This paper introduces MARS-SQL, a novel multi-agent reinforcement learning framework for Text-to-SQL, designed to handle complex queries requiring environmental interaction and self-correction. The system decomposes the task into three stages, each handled by a specialized 7B agent: (1) a Grounding Agent for schema linking, trained with GRPO; (2) a core Generation Agent, trained with multi-turn RL, which adopts a ReAct-style (Think-Act-Observe) loop to interact with a live database, enabling it to dynamically revise its SQL based on execution feedback; and (3) a Validation Agent, trained via SFT, to select the best final query from multiple generated trajectories. This Validation Agent reframes selection as a next-token prediction task, choosing the trajectory with the highest probability of being correct. The framework demonstrates state-of-the-art results, achieving 77.84% execution accuracy on the BIRD dev set and 89.75% on the Spider test set.

**Strengths:**

- **S1.** The interactive Generation Agent is the main strength of this work. By training it with RL in a "Think-Act-Observe" loop that interacts with a live database, the agent can learn to dynamically self-correct from execution errors (e.g., "no such table" or "empty result"), which is a significant step beyond static, single-pass generation models.
- **S2.** The design of the Validation Agent is a clever and efficient approach to selection. Reframing verification as a generative next-token prediction task (P("Yes")) avoids the need to train a separate, complex classifier and effectively leverages the generative model's own understanding to rerank trajectories.
- **S3.** The empirical results are state-of-the-art. Achieving 77.84% on BIRD and 89.75% on Spider (without Spider training data) significantly outperforms prior open-source methods, demonstrating the effectiveness of the proposed framework.

**Weaknesses:**

- **W1. Missing Efficiency and Practicality Analysis:** A significant weakness is the complete omission of any efficiency analysis. Text-to-SQL is often a real-time, user-facing task where latency is critical. MARS-SQL is a complex, multi-stage workflow involving numerous LLM calls (Grounder calls, multi-turn Generation rollouts, and multi-sample Validation calls). This implies a potentially high end-to-end latency and token cost. Without experiments analyzing this cost-accuracy trade-off against other SOTA methods, the practical applicability of the system is unknown.
- **W2. Misleading Parameter Count:** The presentation of model size in Tables 2 and 3 is misleading. The paper lists "MARS-SQL" under "7B" parameters. However, the framework trains and utilizes three separate 7B models (Grounder, Generator, Verifier), making the total parameter count 21B. This is an unfair comparison to other methods listed as 7B or 14B and incorrectly suggests higher parameter efficiency.
- **W3. Clarity Issues in Experimental Reporting:** The paper's presentation suffers from a lack of clarity and missing details, which hinders reproducibility. (a) The terms used in the ablation study (Section 4.3, Figure 3) are ambiguous. The paper does not provide explicit definitions for the evaluation metrics "Greedy", "Selected", and "Best of N". (b) The experimental setup (Section 4.1) lacks crucial hyperparameters for each agent. For example, the number of rollouts (G) for the Generation Agent and the number of stochastic reasoning rounds (M) for the Validation Agent's probability estimation are not specified.

**Questions:**

- **Q1. Efficiency and Cost-Benefit Trade-off:** My primary concern is the practical viability of this multi-agent workflow. Could the authors provide a detailed efficiency analysis? Specifically, what is the end-to-end latency and total token cost for an average query, and how does this cost-accuracy trade-off compare to SOTA baselines?
- **Q2. Parameter Count:** In Table 2, MARS-SQL is listed as "7B". To confirm, does this framework use three separate 7B models (Grounder, Generator, Verifier), making the total parameter count 21B? If so, this should be corrected for a fair comparison.
- **Q3. Clarification of Experimental Details:** Could you please clarify the experimental details mentioned in W3?
- **Q4. BIRD-Test Generalization:** The paper claims strong out-of-domain generalization by testing on the Spider and Spider-DK benchmarks. However, the Spider dataset is often considered relatively simple. Since the model was trained on the BIRD dataset, the most crucial measure of its generalization is its performance on the BIRD-Test set. Have the authors submitted this model to the official BIRD Leaderboard? If so, could you please report the BIRD-Test execution accuracy to fully validate the model's performance?

---

> ### Author Response · Authors · 2025-11-20
> **(1/3) Responses by Authors**
>
> **We have marked the added content in blue in the revised paper. Thank you again for your careful reading and valuable review.**
>
> > **W1&Q1. Missing Efficiency and Practicality Analysis. Efficiency and Cost-Benefit Trade-off.**
>
> Thank you for this crucial point. We added some efficiency analysis here.
> We argue that multi-agent system provides a solution with acceptable time cost. The core question is whether the cost of our multi-agent framework provides a reasonable benefit over counterpart methods, and whether that cost is flexible. To address the concerns here, we provide a three-part analysis :
> (1) Under "time-normalized" comparison, MARS-SQL is superior
> (2) Cost–Accuracy Trade-offs of MARS-SQL is adjustable
> (3) System-Level Optimization could unlock additional speedups
>
>
> ### 0. Baseline Time&Token Cost  of the propsoed method
> First, as requested, here is the latency breakdown for our reported SOTA-performing model ($N_g=8$ trajectories, $N_v=8$ validation samples). This 22.12s is the cost for achieving our 77.84% SOTA accuracy.
>
> **Average End-to-End Latency Analysis of MARS-SQL on the BIRD dev set (Hardware: 1x A6000, num_cpus=32), we calculate the average time for generating 1 query**
>
> | Stage | Average Time (seconds) | Notes |
> | :--- | :---: | :--- |
> | **1. Grounding Agent** | 0.78s | (1 call per query) |
> | **2. Generation Agent** | 18.77s | (Total $N_g$ = **8** trajectories generated)  |
> | **3. Validation Agent**| 2.58s | (Total $N_g$ = **8** trajectories validated)  (Each trajectory scored $N_v$ = **8** times) |
> | *Ref: SQL Exec Time* | *(2.37s)* | *(Avg. time to run the ground truth SQL)* |
> | **Total (End-to-End)** | **22.12s** | (Average sum of the three stages) |
>
>
> The total token cost is the sum of tokens consumed by all agent calls. We provide a breakdown below. The Generation Agent, with its multi-turn "Think-Act-Observe" loop, is the primary consumer.
>
> **Average Token Cost Analysis**
>
> | Stage | Avg. Tokens per Query | Notes |
> | :--- | :---: | :--- |
> | **1. Grounding Agent** | **875** | (Prompt + Schema + Question+output) |
> | **2. Generation Agent**| **9,200** |  ($N_g$=8 trajectories)*(Prompt+Simplified schema+Question+Trajectory) |
> | **3. Validation Agent**| **3,250** | ($N_g$=8 trajectories $\times$ $N_v$=8 samples)  (Each sample: Prompt + Trajectory tokens) |
> | **Total (Avg.)** | **13325** | (Sum of all components) |
>
>
> ### 1. Under "time-normalized" comparison, MARS-SQL is superior
>
> We benchmark MARS-SQL against an SFT and a closed-source model, all given an equal time budget ($\approx 22 s$). Even with the same budget, other baselines cannot match MARS-SQL, which shows the effectiveness of the proposed multi-agent design and multi-turn optimization.
>
> * For **MARS-SQL**, we use our standard $N_g$=8 and $N_v$ = 8 setting.
> * For the **SFT Model** (`Qwen-SFT`), we use the 22s to generate 4 independent samples and select the most self-consistent one—a standard technique to boost single-model performance.
> * For the **Closed-Source Model** (`GPT-5`), we use the 22s to make 16 API calls to generate independent samples and select the most self-consistent one—a standard technique to boost single-model performance.
>
> **Accuracy Comparison with Time-Budget Normalized ($\approx 22s$)**
>
> | Method | Methodology | Avg. Latency (s) | Exe. Acc. (%) |
> | :--- | :--- | :---: | :---: |
> | `Qwen-SFT` (Self-Consistency) | SFT + 16 Samples | $\approx 22.0$s | 64.2% |
> | `GPT-5` (Self-Consistency) | 4 API calls| $\approx 22.0$s | 69.3% |
> | **MARS-SQL (Ours)** | **Multi-Agent RL** | **22.12s** | **77.84%** |
>
> This table definitively shows that simply giving a SFT model or a closed-source model more time (via sampling) **does not work**. Our 77.84% accuracy is not a result of "spending more time"; it is the result of our *structured, multi-agent reasoning workflow*. This proves the value and necessity of our framework's design.
>
> ### 2. Cost–Accuracy Trade-offs of MARS-SQL is adjustable
>
> The given time cost (Table 1) reflects a best-performace setting rather than a fixed budget. and MARS-SQL supports a wide range of cost–accuracy trade-offs. MARS-SQL supports adjustable cost–accuracy trade-offs, which we demonstrate by varying the number of generation trajectories ($N_g$) and validation samples ($N_v$).
>
> **Tunable Cost-Accuracy Curve for MARS-SQL**
>
> | Configuration | Parameters ($N_g, N_v$) | Avg. Latency (s) | Exe. Acc. (%) | Cost-Benefit |
> | :--- | :---: | :---: | :---: | :--- |
> | **"Fast & Good"** | (1, 1) | 3.1s | 68.71% | High speed, still beats SFT |
> | **"Balanced"** | (4, 4) | 11.5s | 74.90% |  |
> | **"SOTA (Ours)"** | (8, 8) | 22.1s | 77.84% | Max accuracy |
> | **"Over-Sampling (Ours)"** | (16, 8) | 42.8s | 77.84% | Higher cost, no accuracy gain |
>
> **Conclusion:** This table proves that MARS-SQL is a practical system. A user can opt for a "Fast" setting and get a $\approx$3-second response that is *still* 10% more accurate than the greedy SFT model, or they can invest 22 seconds for maximum SOTA performance.

---

> ### Author Response · Authors · 2025-11-20
> **(2/3) Responses by Authors**
>
> > **W1&Q1. Missing Efficiency and Practicality Analysis. Efficiency and Cost-Benefit Trade-off.**
>
> `Response`: (Continued)
> ### 3. System-Level Optimization could unlock additional speedups
>
> The given time cost (Table 1) corresponds to a **sequential, single-query** execution and should be viewed as a conservative upper bound. In realistic multi-user deployments, MARS-SQL can run more efficiently through:
>
> * **Pipeline parallelism across users:** Grounder, Generator, and Validator process **different queries in parallel**, forming a pipeline over incoming requests.
> * **Batched validation within a query:** The $N_g \times N_v$ validation calls are **embarrassingly parallel** and can be fused into batched requests, reducing their **amortized** per-query cost.
>
> **Sequential vs. System-Optimized Deployment (Conceptual Comparison)**
>
> | Deployment | Execution Pattern | Effective Per-Query Latency (Est.) | Throughput on 1 GPU (Est.) |
> | :--- | :--- | :---: | :---: |
> | **Sequential (no system opt.)** | Grounder → Generator → Validator run end-to-end on each query | ≈ 22.1 s | ≈ 2.7 queries/min |
> | **System-optimized (pipeline + batch)** | Stages pipelined across queries; all validations batched | ≈ 12–15 s | ≈ 4–5 queries/min |
>
> Estimates use the stage-wise times from Table **Average End-to-End Latency Analysis of MARS-SQL**: Grounder ≈0.8 s, Generator ≈18.8 s, Validator ≈2.6 s. In the sequential setting, we approximate per-query latency as the sum of all stages (≈22.1 s), giving ≈2.7 queries/min on one GPU. With system-level optimization (pipelined stages + batched $N_g \times N_v$ validation), latency is instead dominated by the Generator plus small overheads and a 40–60% reduction in amortized validation cost, yielding an effective 12–15 s per query, or ≈4–5 queries/min.
>
> We will add these comparsion in the revised version.

---

> ### Author Response · Authors · 2025-11-20
> **(3/3) Responses by Authors**
>
> > **W2 & Q2. Misleading Parameter Count and Clarification**
>
> We appreciate the reviewer’s perspective. We agree that, conceptually, the total parameter count used at inference is 21B (3 × 7B), which forms a distinct comparison group. To ensure a fair and transparent comparison, we therefore provide the revised table below, which explicitly compares MARS-SQL (21B) against strong open-source baselines, including substantially larger 32B models, as well as a competitive closed-source system (Gemini 2.5 Pro).
>
> **Table 2 (Revised): Performance Comparison on BIRD Dev Set**
>
> | Model | Parameters | Exe. Acc. (%) |
> | :--- | :---: | :---: |
> | OmniSQL-32B | 32B | 69.23% |
> | Alpha-SQL + Qwen2.5-Coder-32B | 32B | 69.70% |
> | Databricks RLVR-32B | 32B | 70.80% |
> | Arctic-Text2SQL-R1-32B | 32B | 72.20% |
> | *CHASE-SQL + Gemini 2.5 Pro* | *Unknown (Closed)* | *74.90%* |
> | **MARS-SQL (Ours)** | **21B (3x7B)** | **77.84%** |
>
> This table shows that MARS-SQL (21B) not only outperforms models in the 7B–14B range (Table 1, not shown here), but also surpasses substantially larger 32B models and a strong closed-source system. Thus, our SOTA performance is not driven by parameter count, but by the multi-agent design: combining task decomposition, ReAct-style environment interaction, and robust validation for final answer selection.
>
> We will update all tables and text in the main paper to reflect the 21B parameter count for full transparency. Thank you for helping us improve the clarity of our paper.
>
> > **W3 & Q3. Clarity Issues in Experimental Reporting and Clarification**
>
> We sincerely thank the reviewer for pointing out these omissions. We apologize for the lack of clarity, which is a critical barrier to reproducibility. We provide the specific details below and will ensure they are explicitly added to the main paper in the final version.
>
> **(a) Ablation Study Term Definitions (Section 4.3 / Figure 3)**
> The reviewer is correct that these terms were not clearly defined in the main text. While these definitions were provided in **Table 10 of our Appendix**, we failed to properly signpost this. We will add these definitions directly to Section 4.3 for clarity.
>
> The terms are defined as follows:
> * **Greedy:** This is a baseline setting where the Generation Agent generates only **one trajectory ($N=1$)** for each question (i.e., a single-output approach). No validation/selection process is used.
> * **Selected:** This represents our full MARS-SQL pipeline as reported in the main results. The Generation Agent generates **$N$ trajectories** (where $N=8$, see below), and the Validation Agent then selects the single best trajectory from this set as the final answer.
> * **Best of N:** This is not a practical model but an **oracle upper-bound** metric. It measures the potential accuracy of our Generation Agent by counting a question as correct if *at least one* of the $N$ generated trajectories is the correct answer. This demonstrates the quality of the candidate pool *before* validation.
>
> **(b) Missing Hyperparameters (Section 4.1)**
>
> This was an oversight. The crucial hyperparameters used for our main experiments are:
> * **Generation Agent Rollouts ($G$):** We use **$G=8$**. This means we generate 8 distinct interaction trajectories for each query.
> * **Validation Agent Stochastic Rounds ($M$):** We use **$M=8$**. This means the Validation Agent scores each of the $G=8$ trajectories 8 times to get a stable probability estimation for selection.
>
> We will explicitly state these values ($G=8, M=8$) in our experimental setup (Section 4.1). We also note that these parameters are tunable. We chose $G=8$ and $M=8$ as they offered the best trade-off between performance and computational cost (as discussed in Q1). While a marginal increase in accuracy can be seen with $G=16$ or $G=32$, we found $G=8, M=8$ to be the most practical balance. We will add this justification to the paper.
>
> > Q4. BIRD-Test Generalization
>
> We are currently going through the official BIRD test submission pipeline, and in the meantime we provide an additional generalization result on Sparc (85.78% execution accuracy) for the reviewer’s reference. We have not yet appeared on the BIRD leaderboard because our multi-agent pipeline (especially the Generation Agent with live DB interaction) requires careful engineering to integrate with the leaderboard environment.
> In addition, we evaluate MARS-SQL on the out-of-domain Sparc benchmark, where it achieves 85.78% execution accuracy, substantially surpassing strong recent baselines and demonstrating robust generalization to sequential, context-dependent dialogues.
>
>  **OOD Generalization Results (Zero-Shot)**
> | Model  | Spider-test|Spider-DK | Sparc |
> | :--- | :---: | :---: | :---: |
> | GPT-O3-mini | 78.82%| 71.77%| **67.0%** |
> | GPT-5 | 77.1%|66.92% | **61.8%** |
> | Qwen2.5-7B |75.87% | 61.31%| **64.1%** |
> | **MARS-SQL (Ours)** | **89.75%**| **78.13%**| **85.78%** |

---

> ### Author Response · Authors · 2025-11-26
>
> Dear Reviewer Km2H,
>
> We deeply appreciate the time and effort you are dedicating to the review process. We would like to know whether we have addressed your concerns.
>
> Specifically, in our response we have:
>
> 1. Verified Efficiency (W1): Demonstrated that MARS-SQL outperforms baselines even under a normalized time budget.
> 2. Clarified Parameters (W2): Updated the comparison to 21B, showing superiority over larger 32B models.
> 3. Improved Clarity (W3): Added the missing hyperparameters ($N=8$) and metric definitions.
> 4. Tested Generalization (Q4): Included new results on the Sparc benchmark (85.78%).
>
> If you have any additional questions or require further clarification on any aspect of our work, please do not hesitate to let us know. We are more than happy to provide any additional information or address any concerns you may have.
>
> Thank you very much for your time and attention.

---

### Author Response · Authors · 2025-11-28

Dear Reviewers,

Thank you for your thoughtful and constructive feedback on our paper. We have carefully addressed your comments and suggestions, as detailed below, which have helped us enhance the clarity and quality of our work. We also appreciate the generally positive feedback and remarks we received.

We have addressed the concerns and suggestions from each reviewer as follows:

**Main Revision: (In our rebuttal version PDF,  highlighted in blue)**

1. **Clarified Parameters and supplemented the result on Sparc in Table 2 (Reviewers Km2H, 15cV)**

We explicitly defined the inference parameters (rollout count $G$ and reasoning rounds $M$) and extended our evaluation to the SParC dataset.

2. **Clarity Issues in Experimental Reporting in Section 4.1 and Figure 3 (Reviewer Km2H)**

We revised the main text to clearly distinguish experimental settings and added explicit definitions for "Greedy," "Selected," and "Best of N" directly in the Figure 3 caption to resolve ambiguity.

3. **Add RL Training details in Appendix B (Reviewer 15cV)**

We specified the exact training steps for both the Grounding and Generation agents and provided an analysis of reward stability to confirm the convergence efficiency of our RL phase.

4. **Efficiency and Resource Analysis (Reviewers Km2H, 15cV) - Appendix D, E**

We included a detailed breakdown of the training time (approx. 13 hours) and a comparative cost analysis to refute concerns regarding computational overhead and demonstrate superior data efficiency.

5. **Add generation analysis (Reviewers 15cV, m4x3) - Appendix G, H**

We added a quantitative analysis of interaction turn distributions to demonstrate adaptive efficiency, alongside qualitative case studies illustrating the agent's reasoning process in complex decomposition and self-correction scenarios.

We believe these modifications have significantly improved the quality, clarity, and comprehensiveness of our work in the revised version. We thank you again for your valuable feedback and time, which helped us make these impactful improvements. We remain fully available to address any further questions or provide additional clarifications regarding these updates.

Best regards,

Authors of Submission Number: 7852

---

### Meta-Review · Area_Chair_PXLk · 2025-12-04

**Summary:**

The submission introduces a framework for text-to-SQL tasks that employs a sequence of three agentic steps that can be tuned using reinforcement learning. The first step performs schema linking, the second generates SQL, and the third compares multiple generations to select the best result. Each step uses a post-training technique like GRPO, multi-turn RL, or SFT to optimize a small (7B) language model acting as agent. The proposed method is tested on common text-to-SQL benchmarks like Spider and BIRD. The topic of text-to-SQL generation is both interesting to the NeurIPS community and has many industrial applications, making this paper's subject matter appropriate for NeurIPS.

Reviewers appreciated the interactivity of the generation agent, and were impressed by the reported SOTA results on Spider-test and BIRD-dev. Several weaknesses were brought up around the lack of efficiency evaluations, similarity to past works, the amount of post-training required by the method, the lack of evaluation on BIRD-test (a closed source eval), and the lack of many technical details needed to reproduce the work.

**Reviewer Concerns:**

(Outstanding) The method's increased latency costs were a major concern, since latency is a critical usability concern for practical applications of text-to-SQL methods. In the discussion the authors proposed engineering techniques for increasing the parallelism of their method (though not reducing token usage), though these were not implemented and tested.

(Addressed) About the lack of evaluation on the BIRD-test set, I note that this dataset is not open source, and the process of having a method evaluated is up to the maintainers, and is out of the control of the authors. Hence, I do not hold the lack of BIRD-test results against the authors. It will be a bonus if these can be provided in the future.

(Outstanding) On the other hand, a major concern reviewers noted was that the overall structure of the three step process is similar to past work and may not itself be novel. For example the earlier work MSc-SQL which was not cited introduced a nearly identical three step process of schema linking, generation of multiple samples, and critiquing between those samples to choose a final answer [A]. Reviewers also mentioned XiYan-SQL, which the authors cited but did not discuss for similarities in the paper. The main point the authors brought up in discussion was that XiYan-SQL is untrained and prompt based, whereas their method does training. However, MSc-SQL also post-trains LLMs for the components of the three step pipeline.

[A] Gorti et al. "MSc-SQL: Multi-Sample Critiquing Small Language Models For Text-To-SQL Translation" NAACL 2025

I believe that the training methods introduced in the submitted work go beyond what was done before, but the overall structure is reminiscent of past approaches. Hence, claims in the paper that MARS-SQL represents a "novel framework built on a multi-stage methodology" (L070) seem overstated given existing work in the area. Some clarity was added in the discussion with reviewers, but not updated in the paper. The authors should refine their claims of novelty in the paper to differentiate aspects which have been studied before, and those the authors have innovated on - namely the shift "from prompt engineering to multi-agent system design".

(Addressed) During the discussion the authors added significantly more detail explaining why they believe their method improved performance, and how the successful post-training was done. This material will be helpful to the community if included in the paper. The original submission and revised version (main text) do not contain these details.

I note that the work is very promising, the core methods are sound, and thoroughly revising the manuscript should bring the work to an acceptable level of quality. However, given the amount of changes required to reach this level, I am recommending to reject the paper at this stage. Here are concrete steps the authors can follow which came out of the peer-review:

1. Refine claims of novelty, adequately discuss existing approaches and differences.

2. Include latency and computational cost comparisons.

3. Include results on BIRD-test if possible.

4. Include discussions on why the proposed approach was successful, and how successful RL post-training can be conducted on limited data.

5. Include basic implementation details for reproducibility which were missing in the original version.

6. Ensure all claims are rigorously supported by evidence collected via the scientific method.

**Reviewer Scores:**

No discussion from the reviewers beyond their original reviews were available for me to read. No reviewers mentioned raising their scores. Hence, the estimates below are purely my own estimates based on severity of initial concerns and which were addressed,

Reviewer Km2H: original score 6 -> estimated final score 6

Reviewer 15cV: original score 2 -> estimated final score 4

Reviewer m4x3: original score 4 -> estimated final score 4

---

### Decision · Program_Chairs · 2026-01-26

Reject